# The First Optimal Algorithm for Smooth and Strongly-Convex-Strongly-Concave Minimax Optimization

**Dmitry Kovalev**
KAUST[*]
dakovalev1@gmail.com

**Alexander Gasnikov**
MIPT,[†] ISP RAS,[‡] HSE[§]
gasnikov@yandex.ru

## Abstract

In this paper, we revisit the smooth and strongly-convex-strongly-concave minimax optimization problem. Zhang et al. (2021) and Ibrahim et al. (2020) established the lower bound $\Omega\left(\sqrt{\kappa_x \kappa_y} \log \frac{1}{\epsilon}\right)$ on the number of gradient evaluations required to find an $\epsilon$-accurate solution, where $\kappa_x$ and $\kappa_y$ are condition numbers for the strong convexity and strong concavity assumptions. However, the existing state-of-the-art methods do not match this lower bound: algorithms of Lin et al. (2020) and Wang and Li (2020) have gradient evaluation complexity $\mathcal{O}\left(\sqrt{\kappa_x \kappa_y} \log^3 \frac{1}{\epsilon}\right)$ and $\mathcal{O}\left(\sqrt{\kappa_x \kappa_y} \log^3(\kappa_x \kappa_y) \log \frac{1}{\epsilon}\right)$, respectively. We fix this fundamental issue by providing the first algorithm with $\mathcal{O}\left(\sqrt{\kappa_x \kappa_y} \log \frac{1}{\epsilon}\right)$ gradient evaluation complexity. We design our algorithm in three steps: (i) we reformulate the original problem as a minimization problem via *the pointwise conjugate function*; (ii) we apply a specific variant of the proximal point algorithm to the reformulated problem; (iii) we compute the proximal operator inexactly using the optimal algorithm for operator norm reduction in monotone inclusions.

## 1 Introduction

In this paper, we revisit the smooth and strongly-convex-strongly-concave minimax optimization problem of the form

$$\min_{x \in \mathbb{R}^{d_x}} \max_{y \in \mathbb{R}^{d_y}} r(x) + F(x, y) - g(y), \tag{1}$$

where $F(x, y) \colon \mathbb{R}^{d_x} \times \mathbb{R}^{d_y} \to \mathbb{R}$ is a continuously differentiable function, $r(x) \colon \mathbb{R}^{d_x} \to \mathbb{R} \cup \{+\infty\}$ and $g(y) \colon \mathbb{R}^{d_y} \to \mathbb{R} \cup \{+\infty\}$ are proper lower semi-continuous convex functions. Problem (1) has been actively studied in economics, game theory, statistics and computer science (Başar and Olsder, 1998; Roughgarden, 2010; Von Neumann and Morgenstern, 1947; Facchinei and Pang, 2003; Berger, 2013). Recently, many applications of this problem appeared in machine learning, including adversarial training (Madry et al., 2017; Sinha et al., 2017), prediction and regression problems (Taskar et al., 2005; Xu et al., 2009), reinforcement learning (Du et al., 2017; Dai et al., 2018) and generative adversarial networks Arjovsky et al. (2017); Goodfellow et al. (2014).

In our paper, we focus on the case when function $f(x, y)$ is strongly convex in $x$ and strongly concave in $y$. There are several reasons to consider this function class. First, this setting is fundamental and

---

[*]King Abdullah University of Science and Technology, Thuwal, Saudi Arabia

[†]Moscow Institute of Physics and Technology, Dolgoprudny, Russia

[‡]Institute for System Programming RAS, Research Center for Trusted Artificial Intelligence, Moscow, Russia

[§]National Research University Higher School of Economics, Moscow, Russia

36th Conference on Neural Information Processing Systems (NeurIPS 2022).

Table 1: Comparison of the state-of-the-art algorithms for solving smooth and strongly-convex-strongly-concave minimax problems in the number of gradient evaluations required to find an $\epsilon$-accurate solution (Definition 1).

| Reference | Gradient Complexity |
|:---:|:---:|
| Tseng (2000) | $\mathcal{O}\left(\max\left\{\kappa_x, \kappa_y\right\} \log \frac{1}{\epsilon}\right)$ |
| Nesterov and Scrimali (2006) | |
| Gidel et al. (2018) | |
| Alkousa et al. (2019) | $\mathcal{O}\left(\min\left\{\kappa_x\sqrt{\kappa_y}, \kappa_y\sqrt{\kappa_x}\right\} \log^2 \frac{1}{\epsilon}\right)$ |
| Lin et al. (2020) | $\mathcal{O}\left(\sqrt{\kappa_x\kappa_y} \log^3 \frac{1}{\epsilon}\right)$ |
| Wang and Li (2020) | $\mathcal{O}\left(\sqrt{\kappa_x\kappa_y} \log^3(\kappa_x\kappa_y) \log \frac{1}{\epsilon}\right)$ |
| **Algorithm 4 (This paper)** | $\mathcal{O}\left(\sqrt{\kappa_x\kappa_y} \log \frac{1}{\epsilon}\right)$ |
| Lower Bound (Zhang et al., 2021; Ibrahim et al., 2020) | $\Omega\left(\sqrt{\kappa_x\kappa_y} \log \frac{1}{\epsilon}\right)$ |

studied by most existing works on minimax optimization.[5] Second, efficient algorithms initially developed for convex optimization often show state-of-the-art performance in non-convex applications (Kingma and Ba, 2014; Reddi et al., 2019; Duchi et al., 2011). Finally, we will further see that this fundamental setting is utterly understudied and lacks answers to even the most basic questions such as "What is the best possible algorithm for solving a problem in this setting?"[6]

## 1.1 Related Work

Until recently, the best-known gradient evaluation complexity of solving problem (1) was $\mathcal{O}\left(\max\left\{\kappa_x, \kappa_y\right\} \log \frac{1}{\epsilon}\right)$ (Tseng, 2000; Nesterov and Scrimali, 2006; Gidel et al., 2018), where $\kappa_x$ and $\kappa_y$ denote the condition numbers of functions $f(\cdot, y)$ and $f(x, \cdot)$, respectively. The first attempt to provide an algorithm with an "accelerated" convergence rate was the work of Alkousa et al. (2019). They provided an algorithm with $\mathcal{O}\left(\min\left\{\kappa_x\sqrt{\kappa_y}, \kappa_y\sqrt{\kappa_x}\right\} \log^2 \frac{1}{\epsilon}\right)$ gradient evaluation complexity. This result was subsequently improved up to $\mathcal{O}\left(\sqrt{\kappa_x\kappa_y} \log^3 \frac{1}{\epsilon}\right)$ by Lin et al. (2020) and $\mathcal{O}\left(\sqrt{\kappa_x\kappa_y} \log^3(\kappa_x\kappa_y) \log \frac{1}{\epsilon}\right)$ by Wang and Li (2020). However, these results do not match the lower complexity bound $\Omega\left(\sqrt{\kappa_x\kappa_y} \log \frac{1}{\epsilon}\right)$ established by Zhang et al. (2021); Ibrahim et al. (2020). Hence, we have the following fundamental open problem:

*Can we design an algorithm that achieves the lower gradient evaluation complexity bound in smooth and strongly-convex-strongly-concave minimax optimization?*

It is worth mentioning that this open question was answered positively in the works of Kovalev et al. (2021); Thekumparampil et al. (2022); Jin et al. (2022) in the case of minimax problems with bilinear coupling, i.e., when $F(x, y) = p(x) + x^\top \mathbf{A} y - q(y)$, where $p(x)$ and $q(y)$ are smooth and strongly convex functions, and $\mathbf{A}$ is a $d_x \times d_y$ matrix. However, the algorithm provided in this work does not apply to the general minimax problem (1).

## 1.2 Main Contributions

We develop the first optimal algorithm for solving problem (1) in the smooth and strongly-convex-strongly-concave regime, which is the main contribution of this work. We split the algorithm development in three steps:

    **(i)** In Section 3, we reformulate problem (1) as a particular minimization problem.

---

[5]Most existing works on minimax optimization study the convex-concave case. However, this setting can be easily reduced to the strongly-convex-strongly-concave case via the regularization technique (Lin et al., 2020).

[6]In contrast to smooth convex-concave minimax optimization, the answer to this question for smooth convex minimization was given by Nesterov (1983) several decades ago.

**(ii)** In Section 4, we develop a specific variant of the accelerated proximal point algorithm (Algorithm 2) which will be used as a baseline for the optimal algorithm construction.

**(ii)** In Section 5, we develop an optimal algorithm for operator norm reduction in monotone inclusion problems, which will be used for the proximal operator computation in Algorithm 2.

In the final Section 6, we summarize these three steps by describing the optimal algorithm construction and showing that the complexity of the proposed algorithm matches the lower bound.

As mentioned before, in Section 5, we develop an optimal algorithm for operator norm reduction in composite monotone inclusion problems of the form (22), which is the second main contribution of this work. To the best of our knowledge, there is only one optimal algorithm of Yoon and Ryu (2021), which works for Lipschitz-continuous operators only, i.e., when $B(u) \equiv 0$ in problem (22). In contrast to this, our algorithm works in the composite case with general maximally monotone operator $B(u)$.

**Concurrent work of Carmon et al. (2022).** In their recent concurrent paper, Carmon et al. (2022) developed an efficient variant of the Catalyst method (Lin et al., 2015) called RECAPP. In contrast to the original Catalyst method, RECAPP does not suffer from extra logarithmic factors in the complexity. When applied to the minimax optimization problem (1), RECAPP achieves the optimal complexity $\mathcal{O}\left(\sqrt{\kappa_x \kappa_y} \log \frac{1}{\epsilon}\right)$. Hence, RECAPP is another optimal algorithm for solving smooth strongly-convex-strongly-concave minimax optimization problems. However, the work of Carmon et al. (2022) appeared on arXiv and was published at ICML 2022 later than the first version of this paper appeared on arXiv.

## 2 Preliminaries

The following assumptions formalize the smoothness, strong convexity, and strong concavity properties of function $f(x, y)$.

**Assumption 1.** *Function $F(x, y)$ is $\mu_x$-strongly convex in $x$, where $\mu_x > 0$. That is, the following inequality holds for all $x_1, x_2 \in \mathbb{R}^{d_x}, y \in \mathbb{R}^{d_y}$:*

$$F(x_2, y) \geq F(x_1, y) + \langle \nabla_x F(x_1, y), x_2 - x_1 \rangle + (\mu_x/2)\|x_2 - x_1\|^2. \tag{2}$$

**Assumption 2.** *Function $F(x, y)$ is $\mu_y$-strongly concave in $y$, where $\mu_y > 0$. That is, the following inequality holds for all $x \in \mathbb{R}^{d_x}, y_1, y_2 \in \mathbb{R}^{d_y}$:*

$$F(x, y_2) \leq F(x, y_1) + \langle \nabla_y F(x, y_1), y_2 - y_1 \rangle - (\mu_y/2)\|y_2 - y_1\|^2. \tag{3}$$

**Assumption 3.** *Function $F(x, y)$ is $L$-smooth. That is, the following inequality holds for all $x_1, x_2 \in \mathbb{R}^{d_x}, y_1, y_2 \in \mathbb{R}^{d_y}$:*

$$\|\nabla F(x_1, y_1) - \nabla F(x_2, y_2)\|^2 \leq L^2 \left(\|x_1 - x_2\|^2 + \|y_1 - y_2\|^2\right). \tag{4}$$

Under these assumptions, by $\kappa_x = \frac{L}{\mu_x}$ and $\kappa_y = \frac{L}{\mu_x}$, we denote the condition numbers of functions $F(\cdot, y)$ and $F(x, \cdot)$, respectively. The following assumption formalizes the properties of regularizers $r(x)$ and $g(y)$.

**Assumption 4.** *Functions $r(x)$ and $g(y)$ are convex, lower semi-continuous and proper, i.e., there exist $\bar{x} \in \mathbb{R}^{d_x}, \bar{y} \in \mathbb{R}^{d_y}$ such that $r(\bar{x}), g(\bar{y}) < +\infty$.*

By $(x^*, y^*) \in \mathbb{R}^{d_x} \times \mathbb{R}^{d_y}$, we denote the solution of problem (1), which is characterized via the first-order optimality conditions

$$\begin{cases} -\nabla_x F(x^*, y^*) \in \partial r(x^*), \\ \nabla_y F(x^*, y^*) \in \partial g(y^*). \end{cases} \tag{5}$$

Note that there exists a unique solution to the problem due to the strong convexity and strong concavity assumptions (Assumptions 1 and 2). Hence, for any point $(x, y) \in R^{d_x} \times \mathbb{R}^{d_y}$, we can use squared distance to the solution $\|x - x^*\|^2 + \|y - y^*\|^2$ as an optimality criterion. We formalize it through the following definition.

**Definition 1.** *We call a pair of vectors $(x, y) \in \mathbb{R}^{d_x} \times \mathbb{R}^{d_y}$ an $\epsilon$-accurate solution of problem* (1) *for a given accuracy $\epsilon > 0$ if it satisfies*

$$\|x - x^*\|^2 + \|y - y^*\|^2 \leq \epsilon. \tag{6}$$

# 3 Step I: Reformulation via Pointwise Conjugate Function

In this section, we reformulate problem (1) as a particular convex minimization problem. This reformulation will be beneficial because minimization problems are typically easier to solve than minimax optimization problems.

## 3.1 Pointwise Conjugate Function

We start by introducing *the pointwise conjugate function* which will be the main component of our problem reformulation. Let function $\hat{F}(x, y) \colon \mathbb{R}^{d_x} \times \mathbb{R}^{d_y} \to \mathbb{R}$ be defined as

$$\hat{F}(x, y) = F(x, y) - \frac{\mu_x}{2}\|x\|^2 + \frac{\mu_y}{2}\|y\|^2. \tag{7}$$

One can observe that function $\hat{F}(x, y)$ is smooth, convex in $x$, and concave in $y$ due to Assumptions 1 to 3. Now, the pointwise conjugate function $G(z, y) \colon \mathbb{R}^{d_x} \times \mathbb{R}^{d_y} \to \mathbb{R}$ is defined as follows:

$$G(z, y) = \sup_{x \in \mathbb{R}^{d_x}} \left[ \langle x, z \rangle - r(x) - \hat{F}(x, y) + g(y) \right]. \tag{8}$$

One can observe that for fixed $y \in \mathbb{R}^{d_y}$, function $G(\cdot, y)$ is nothing else but the Fenchel conjugate[7] of function $r(\cdot) + \hat{F}(\cdot, y) - g(y)$. Moreover, function $G(z, y)$ is defined as a pointwise supremum of a family of convex and lower semi-continuous functions $\left\{ \varphi_x(z, y) = \langle x, z \rangle - r(x) - F(x, y) + g(y) \mid x \in \mathbb{R}^{d_x} \right\}$. Hence, $G(z, y)$ is also convex and lower semi-continuous function. The following lemma provides a characterization of the subdifferential of the pointwise conjugate function.

**Lemma 1.** *Let $z, x \in \mathbb{R}^{d_x}$ and $y, w \in \mathbb{R}^{d_y}$ be arbitrary vectors that satisfy*

$$z - \nabla_x \hat{F}(x, y) \in \partial r(x), \ w + \nabla_y \hat{F}(x, y) \in \partial g(y). \tag{9}$$

*Then, $G(z, y) = \langle z, x \rangle - r(x) - \hat{F}(x, y) + g(y)$ and $(x, w) \in \partial G(z, y)$.*

## 3.2 Reformulation of the Minimax Optimization Problem

Now, we introduce the following minimization problem:

$$\min_{z \in \mathbb{R}^{d_x}, y \in \mathbb{R}^{d_y}} \left[ P(z, y) = \frac{\mu_x^{-1}}{2}\|z\|^2 + \frac{\mu_y}{2}\|y\|^2 + G(z, y) \right] \tag{10}$$

It turns out that this minimization problem can be seen as a reformulation of problem (1). This is justified by the following lemma.

**Lemma 2.** *Problem* (10) *has a unique solution $(z^*, y^*) \in \mathbb{R}^{d_x} \times \mathbb{R}^{d_y}$, where*

$$z^* = -\mu_x x^* \tag{11}$$

*and $(x^*, y^*)$ is the unique solution of problem* (1).

Lemma 2 implies that if we find an approximate solution $(z, y) \in \mathbb{R}^{d_x} \times \mathbb{R}^{d_y}$ to problem (10), a pair of vectors $(-\mu_x^{-1} z, y) \in \mathbb{R}^{d_x} \times \mathbb{R}^{d_y}$ will be an approximate solution to the original minimax problem.

The idea of reformulating the minimax optimization problem as a minimization problem is not new and has been used in the state-of-the-art works of Lin et al. (2020); Wang and Li (2020); Alkousa et al. (2019). However, their reformulation is different from ours and has several disadvantages. In particular, it does not allow for building the optimal algorithm for solving problem (1). We provide a detailed discussion of this in the Appendix.

# 4 Step II: Accelerated Proximal Point Method

In this section, we develop the main algorithmic framework for solving problem (10), which is formalized as Algorithm 2. We give the intuition behind the development of Algorithm 2 and provide its theoretical analysis. Further, in Section 6, we will use this algorithmic framework to develop the first optimal algorithm for solving main problem (1).

---

[7]Recall that for a convex function $h(x)$, Fenchel conjugate is defined as $h^*(z) = \sup_x [\langle z, x \rangle - h(x)]$.

---

**Algorithm 1** Accelerated Gradient Method

---

1: **input:** $z^0 = z_f^0 \in \mathbb{R}^{d_x}, y^0 = y_f^0 \in \mathbb{R}^{d_y}$
2: **parameters:** $\alpha \in (0,1], \eta_z, \eta_y, \theta_z, \theta_y > 0, K \in \{1, 2, \ldots\}$
3: **for** $k = 0, 1, 2, \ldots, K - 1$ **do**
4:      $(z_g^k, y_g^k) = \alpha(z^k, y^k) + (1 - \alpha)(z_f^k, y_f^k)$
5:      $z_f^{k+1} = z_g^k - \theta_z \nabla_z P(z_g^k, y_g^k)$
6:      $y_f^{k+1} = y_g^k - \theta_y \nabla_y P(z_g^k, y_g^k)$
7:      $z^{k+1} = z^k + \eta_z \mu_z(z_g^k - z^k) + \eta_z \theta_z^{-1}(z_f^{k+1} - z_g^k)$
8:      $y^{k+1} = y^k + \eta_y \mu_y(y_g^k - y^k) + \eta_y \theta_y^{-1}(y_f^{k+1} - y_g^k)$
9: **end for**
10: **output:** $(z^K, y^K)$

---

---

**Algorithm 2** Accelerated Proximal Point Algorithm

---

1: **input:** $z^0 = z_f^0 \in \mathbb{R}^{d_x}, y^0 = y_f^0 \in \mathbb{R}^{d_y}$
2: **parameters:** $\alpha \in (0,1], \eta_z, \eta_y, \theta_y > 0, K \in \{1, 2, \ldots\}$
3: **for** $k = 0, 1, 2, \ldots, K - 1$ **do**
4:      $(z_g^k, y_g^k) = \alpha(z^k, y^k) + (1 - \alpha)(z_f^k, y_f^k)$
5:      Find $(x_f^{k+1}, y_f^{k+1}, z_f^{k+1}, w_f^{k+1}) \in \mathbb{R}^{d_x} \times \mathbb{R}^{d_y} \times \mathbb{R}^{d_x} \times \mathbb{R}^{d_y}$ that satisfy (18)
6:      $z^{k+1} = z^k + \eta_z \mu_x^{-1}(z_f^{k+1} - z^k) - \eta_z(x_f^{k+1} + \mu_x^{-1} z_f^{k+1})$
7:      $y^{k+1} = y^k + \eta_y \mu_y(y_f^{k+1} - y^k) - \eta_y(w_f^{k+1} + \mu_y y_f^{k+1})$
8: **end for**
9: **output:** $(z^K, y^K)$

---

## 4.1 Nesterov Acceleration

It is well-known that Accelerated Gradient Method of Nesterov (1983, 2003) is the optimal algorithm for solving smooth (strongly-)convex minimization problems. Therefore, we could try to apply this method to solving problem (10), which is formalized as Algorithm 1. Note that we used the notation $\mu_z = \mu_x^{-1}$ in Algorithm 1, which is the strong convexity parameter of $P(z, y)$ in $z$. Unfortunately, function $P(z, y)$ can be non-smooth, and the gradient $\nabla P(z_g^k, y_g^k)$ can be undefined. It means that Algorithm 1 cannot be applied to problem (10).

## 4.2 Moreau-Yosida Regularization

In order to avoid the issues caused by the non-smoothness of function $P(z, y)$, we use the Moreau-Yosida regularization (Moreau, 1962; Yosida, 2012). Consider a function $P^{\theta_z, \theta_y}(z, y)$ defined in the following way:

$$P^{\theta_z, \theta_y}(z, y) = \min_{z^+ \in \mathbb{R}^{d_x}, y^+ \in \mathbb{R}^{d_y}} \frac{1}{2\theta_z} \|z^+ - z\|^2 + \frac{1}{2\theta_y} \|y^+ - y\|^2 + P(z^+, y^+), \qquad (12)$$

where $\theta_z, \theta_y > 0$. Function $P^{\theta_z, \theta_y}(z, y)$ is called the Moreau envelope of function $P(z, y)$. The Moreau envelope has two crucial properties. First, it is a smooth function. Second, it has the same minimizers as function $P(z, y)$:

$$(z^*, y^*) = \underset{z \in \mathbb{R}^{d_x}, y \in \mathbb{R}^{d_y}}{\arg\min} P^{\theta_z, \theta_y}(z, y). \qquad (13)$$

The latter means that we could apply Accelerated Gradient Method to problem (13), which would give us an efficient algorithm for solving problem (10). Further, we are going to construct such an algorithm.

## 4.3 Construction of the Algorithm

We start the construction of our algorithm by computing the gradient $\nabla P^{\theta_z, \theta_y}(z_g^k, y_g^k)$. The theory of the Moreau-Yosida regularization (Lemaréchal and Sagastizábal, 1997) suggests that the gradient of

the Moreau envelope can be computed in the following way:

$$\nabla P^{\theta_z, \theta_y}(z_g^k, y_g^k) = \begin{bmatrix} \theta_z^{-1}(z_g^k - z_f^{k+1}) \\ \theta_y^{-1}(y_g^k - y_f^{k+1}) \end{bmatrix}, \tag{14}$$

where $(z_f^{k+1}, y_f^{k+1}) \in \mathbb{R}^{d_x} \times \mathbb{R}^{d_y}$ is computed via the following auxiliary minimization problem:

$$(z_f^{k+1}, y_f^{k+1}) = \underset{z \in \mathbb{R}^{d_x}, y \in \mathbb{R}^{d_y}}{\arg\min} \frac{1}{2\theta_z} \|z - z_g^k\|^2 + \frac{1}{2\theta_y} \|y - y_g^k\|^2 + P(z, y). \tag{15}$$

Further, we choose parameter $\theta_z = \mu_z^{-1} = \mu_x$ and write the first-order optimality conditions for this problem using the definition of function $P(z, y)$:

$$\begin{bmatrix} \mu_x^{-1}(z_f^{k+1} - z_g^k) + \mu_x^{-1} z_f^{k+1} \\ \theta_y^{-1}(y_f^{k+1} - y_g^k) + \mu_y y_f^{k+1} \end{bmatrix} \in -\partial G(z_f^{k+1}, y_f^{k+1}). \tag{16}$$

The latter condition involves the subdifferential $\partial G(z, y)$. Hence, we can rewrite this condition using Lemma 1, which provides the characterization of $\partial G(z, y)$[8]:

$$\begin{aligned} z_f^{k+1} - \nabla_x \hat{F}(x_f^{k+1}, y_f^{k+1}) \in \partial r(x_f^{k+1}), \qquad x_f^{k+1} + \mu_x^{-1}(z_f^{k+1} - z_g^k) + \mu_x^{-1} z_f^{k+1} = 0, \\ w_f^{k+1} + \nabla_y \hat{F}(x_f^{k+1}, y_f^{k+1}) \in \partial g(y_f^{k+1}), \qquad w_f^{k+1} + \theta_y^{-1}(y_f^{k+1} - y_g^k) + \mu_y y_f^{k+1} = 0. \end{aligned} \tag{17}$$

where $x_f^{k+1} \in \mathbb{R}^{d_x}$ and $w_f^{k+1} \in \mathbb{R}^{d_y}$ are auxiliary vectors. From (17) we get

$$\mu_x^{-1}(z_f^{k+1} - z_g^k) = -(x_f^{k+1} + \mu_x^{-1} z_f^{k+1}),$$
$$\theta_y^{-1}(y_f^{k+1} - y_g^k) = -(w_f^{k+1} + \mu_y y_f^{k+1}),$$

which we plug into lines 7 and 8 of Algorithm 1.

Finally, we replace the computation of $(z_f^{k+1}, y_f^{k+1})$ on lines 5 and 6 of Algorithm 1 using condition (17). It turns out that we can use the following relaxed version of condition (17) without hurting the convergence properties of the resulting algorithm:

$$\begin{cases} z_f^{k+1} - \nabla_x \hat{F}(x_f^{k+1}, y_f^{k+1}) \in \partial r(x_f^{k+1}), \\ w_f^{k+1} + \nabla_y \hat{F}(x_f^{k+1}, y_f^{k+1}) \in \partial g(y_f^{k+1}), \\ \dfrac{8}{\mu_x} \|\Delta_x^k\|^2 + \theta_y \|\Delta_y^k\|^2 \le \dfrac{\mu_x}{8} \|x_f^{k+1} + \mu_x^{-1} z_g^k\|^2 + \theta_y^{-1} \|y_f^{k+1} - y_g^k\|^2, \end{cases} \tag{18}$$

where $\Delta_x^k$ and $\Delta_y^k$ are defined as follows:

$$\begin{cases} \Delta_x^k = z_f^{k+1} + \dfrac{\mu_x}{2}(x_f^{k+1} - \mu_x^{-1} z_g^k), \\ \Delta_y^k = w_f^{k+1} + \mu_y y_f^{k+1} + \theta_y^{-1}(y_f^{k+1} - y_g^k). \end{cases} \tag{19}$$

### 4.4 Convergence of the Algorithm

After applying all the modifications mentioned above to Algorithm 1, we obtain Algorithm 2. Theorem 1 provides the iteration complexity of Algorithm 2. The proof of Theorem 1 can be found in the Appendix.

**Theorem 1.** *Let $\eta_z, \eta_y$ be defined as*

$$\eta_z = \mu_x/2, \quad \eta_y = \min\{1/(2\mu_y), \theta_y/(2\alpha)\}. \tag{20}$$

*Then, to find an $\epsilon$-accurate solution of problem (1), Algorithm 2 requires the following number of iterations:*

$$K = \mathcal{O}\left(\max\left\{\frac{1}{\alpha}, \frac{\alpha}{\theta_y \mu_y}\right\} \log \frac{1}{\epsilon}\right). \tag{21}$$

*In this case, the $\epsilon$-accurate solution will be given as $(-\mu_x^{-1} z^K, y^K)$, where $(z^K, y^K)$ is the output of Algorithm 2.*

---

[8]To be precise, Lemma 1 implies the relation (17) $\Rightarrow$ (16) rather than the equivalence (17) $\Leftrightarrow$ (16). However, this is not an issue because we provide the intuition behind the algorithm development in this section. The rigorous proofs are postponed to the Appendix.

---
**Algorithm 3** Extra Anchored Gradient for Monotone Inclusions
---
1: **input:** $u^{-1} \in \mathbb{R}^d$
2: **parameters:** $\lambda > 0$, $T \in \{1, 2, \ldots\}$, $\{\beta_t\}_{t=0}^{T-1} \subset (0, 1)$
3: $u^0 = \mathrm{J}_{\lambda B}(u^{-1} - \lambda A(u^{-1}))$
4: $a^0 = A(u^0)$
5: $b^0 = \frac{1}{\lambda}(u^{-1} - \lambda A(u^{-1}) - u^0)$          $\triangleright b^0 \in B(u^0)$
6: **for** $t = 0, 1, 2 \ldots, T - 1$ **do**
7:     $u^{t+1/2} = u^t + \beta_t(u^0 - u^t) - \lambda(a^t + b^t)$
8:     $u^{t+1} = \mathrm{J}_{\lambda B}(u^t + \beta_t(u^0 - u^t) - \lambda A(u^{t+1/2}))$
9:     $a^{t+1} = A(u^{t+1})$
10:    $b^{t+1} = \frac{1}{\lambda}(u^t + \beta_t(u^0 - u^t) - \lambda A(u^{t+1/2}) - u^{t+1})$     $\triangleright b^{t+1} \in B(u^{t+1})$
11: **end for**
12: **output:** $(u^T, a^T + b^T)$               $\triangleright b^T \in B(u^T)$
---

Unfortunately, Algorithm 2 cannot be applied to solving problem (1) in its current form because it requires finding vectors $(x_f^{k+1}, y_f^{k+1}, z_f^{k+1}, w_f^{k+1})$ that satisfy condition (18) on line 5 at each iteration. Further, we will show that finding these vectors can be seen as finding an approximate solution to a particular monotone inclusion problem. In Section 5, we will provide an optimal algorithm for solving such monotone inclusions. In Section 6, we will show how to combine this algorithm with Algorithm 2 and obtain the first optimal algorithm for solving main problem (1).

## 5 Step III: Operator Norm Reduction in Monotone Inclusions

In this section, we consider the following monotone inclusion problem:

$$\text{find } u^* \in \mathbb{R}^d \quad \text{such that} \quad 0 \in A(u^*) + B(u^*), \tag{22}$$

where $A(u), B(u)\colon \mathbb{R}^d \rightrightarrows \mathbb{R}^d$ are maximally monotone mappings. We are interested in the case when $A(u)$ is single-valued and Lipschitz continuous. The properties of operators $A(u)$ and $B(u)$ are formalized through the following assumptions.

**Assumption 5.** *Mapping $A(u)\colon \mathbb{R}^d \to \mathbb{R}^d$ is single-valued, $M$-Lipschitz and monotone. That is, for all $u_1, u_2 \in \mathbb{R}^d$, $\langle A(u_1) - A(u_2), u_1 - u_2 \rangle \geq 0$ and $\|A(u_1) - A(u_2)\| \leq M\|u_1 - u_2\|$.*

**Assumption 6.** *Mapping $B(u)\colon \mathbb{R}^d \rightrightarrows \mathbb{R}^d$ is maximally monotone and possibly multivalued. That is, mapping $B(u)$ satisfies the following conditions:*

    *1. $B(u)$ is monotone, i.e., for all $u_1, u_2 \in \mathrm{dom}\, B$, $b_1 \in B(u_1)$, $b_2 \in B(u_2)$ the following inequality holds: $\langle u_1 - u_2, b_1 - b_2 \rangle \geq 0$, where $\mathrm{dom}\, B = \{u \in \mathbb{R}^d \mid B(u) \neq \emptyset\}$.*

    *2. The graph $\mathrm{gph}\, B = \{(u, b) \in \mathbb{R}^d \times \mathbb{R}^d \mid b \in B(u)\}$ is not properly contained in the graph of any other monotone mapping on $\mathbb{R}^d$.*

Note that mapping $A(u)$ is also maximally monotone because it is monotone and continuous (Rockafellar and Wets, 2009, Example 12.7). Further, we will use an operator $\mathrm{J}_{\lambda B}(u)\colon \mathbb{R}^d \rightrightarrows \mathbb{R}^d$ which is defined as

$$u^+ \in \mathrm{J}_{\lambda B}(u) \quad \text{if and only if} \quad \lambda^{-1}(u - u^+) \in B(u^+), \tag{23}$$

where $\lambda > 0$. This mapping is called the resolvent of mapping $B(u)$. The maximal monotonicity of $B(u)$ implies that the resolvent $\mathrm{J}_{\lambda B}(u)$ is single-valued for all $u \in \mathbb{R}^d$ (Rockafellar and Wets, 2009, Theorem 12.12).

Now, we are ready to present Algorithm 3 for solving monotone inclusion problem (22). The design of our algorithm is based on the Extra Anchored Gradient Algorithm of Yoon and Ryu (2021). The critical difference between the algorithm of Yoon and Ryu (2021) and Algorithm 3 is that the algorithm of Yoon and Ryu (2021) can be applied to problem (22) in the case $B(u) \equiv \{0\}$ only. Therefore, our Algorithm 3 can be seen as an extension of the algorithm of Yoon and Ryu (2021) for general monotone inclusion problems of the form (22).

The following theorem provides the convergence guarantees for Algorithm 3. The proof of the theorem can be found in the Appendix.

**Theorem 2.** *Assume that there exists at least a single solution $u^*$ to problem* (22). *Let $\beta_t$ be defined as follows*

$$\beta_t = 2/(t+3). \tag{24}$$

*Let $\lambda$ be defined as*

$$\lambda = 1/(\sqrt{5}M). \tag{25}$$

*Then, the following inequality holds*

$$\|a^T + b^T\|^2 \le \frac{288M^2}{(T+1)^2}\|u^{-1} - u^*\|^2, \tag{26}$$

*where $a^T = A(u^T)$ and $b^T \in B(u^T)$.*

## 6 Final Step: The First Optimal Algorithm for Minimax Optimization

In this section, we construct the first optimal algorithm for solving main problem (1). In order to do this, we use Algorithm 3 to compute vectors $(x_f^{k+1}, y_f^{k+1}, z_f^{k+1}, w_f^{k+1})$ on line 5 of Algorithm 2. Further, we describe the construction of our algorithm in detail.

### 6.1 Construction of the Algorithm

As mentioned in Section 4, Algorithm 2 cannot be applied to solving problem (1) in its current form because it requires finding the vectors satisfying condition (18) on line 5 at each iteration. Further, we will show how to do this using Algorithm 3. Let $\mathbb{R}^d = \mathbb{R}^{d_x} \times \mathbb{R}^{d_y}$. For each $k \in \{0, 1, 2, \ldots\}$ consider operators $A^k(u)\colon \mathbb{R}^d \to \mathbb{R}^d$ and $B(u)\colon \mathbb{R}^d \rightrightarrows \mathbb{R}^d$ defined as follows:

$$A^k(u) = \begin{bmatrix} \sqrt{\gamma_x}a_x^k(x,y) \\ \sqrt{\gamma_y}a_y^k(x,y) \end{bmatrix}, \quad B(u) = \left\{ \begin{bmatrix} \sqrt{\gamma_x}b_x \\ \sqrt{\gamma_y}b_y \end{bmatrix} \,\middle|\, b_x \in \partial r(x), b_y \in \partial g(y) \right\}, \tag{27}$$

where $\gamma_x, \gamma_y > 0$ are parameters, variable $u \in \mathbb{R}^d$ is defined as

$$u = (\gamma_x^{-1/2}x, \gamma_y^{-1/2}y), \quad \text{where} \quad (x,y) \in \mathbb{R}^{d_x} \times \mathbb{R}^{d_y}, \tag{28}$$

and operators $a_x^k(x,y)\colon \mathbb{R}^d \to \mathbb{R}^{d_x}$ and $a_y^k(x,y)\colon \mathbb{R}^d \to \mathbb{R}^{d_y}$ are defined as

$$\begin{aligned} a_x^k(x,y) &= \nabla_x \hat{F}(x,y) + \frac{\mu_x}{2}(x - \mu_x^{-1}z_g^k), \\ a_y^k(x,y) &= -\nabla_y \hat{F}(x,y) + \mu_y y + \theta_y^{-1}(y - y_g^k). \end{aligned} \tag{29}$$

One can observe that operators $A^k(u)$ and $B(u)$ satisfy Assumptions 5 and 6. This is justified by the following lemma.

**Lemma 3.** *Operator $A^k(u)$, defined by* (27), *is monotone and $M$-Lipschitz, where $M$ is given as*

$$M = 2\max\{\gamma_x L, \gamma_y(L + \theta_y^{-1})\}. \tag{30}$$

*Operator $B(u)$, defined by* (27), *is maximally monotone.*

Now, we are ready to construct the first optimal algorithm for solving main problem (1) which is formalized as Algorithm 4 (it can be found in the Appendix). In order to do this, we use Algorithm 3 to perform the computations on line 5 of Algorithm 2. Consider the $k$-th iteration of Algorithm 2 and replace line 5 of Algorithm 2 with the lines of Algorithm 3 using the notation $u^t = (\gamma_x^{-1/2}x^{k,t}, \gamma_y^{-1/2}y^{k,t})$ for $t \in \{-1, 0, 1, 2, \ldots\}$.

In addition, we replace the for-loop of Algorithm 3 with the while-loop that iterates until the following condition is satisfied (see line 11 of Algorithm 4):

$$\begin{aligned} \gamma_x \|a_x^k(x^{k,t}, y^{k,t}) + b_x^{k,t}\|^2 + \gamma_y \|a_y^k(x^{k,t}, y^{k,t}) + b_y^{k,t}\|^2 &\le \\ \le \gamma_x^{-1}\|x^{k,t} - x^{k,-1}\|^2 &+ \gamma_y^{-1}\|y^{k,t} - y^{k,-1}\|^2. \end{aligned} \tag{31}$$

---
**Algorithm 4** FOAM: The First Optimal Algorithm for Minimax Optimization
---
1: **input:** $z^0 = z_f^0 \in \mathbb{R}^{d_x}$, $y^0 = y_f^0 \in \mathbb{R}^{d_y}$
2: **parameters:** $\alpha \in (0,1]$, $\eta_z, \eta_y, \theta_y > 0$, $\{\beta_t\}_{t=0}^{\infty} \subset (0,1)$, $\lambda, \gamma_x, \gamma_y > 0$, $K \in \{1, 2, \ldots\}$
3: **for** $k = 0, 1, 2, \ldots, K - 1$ **do**
4:     $(z_g^k, y_g^k) = \alpha(z^k, y^k) + (1 - \alpha)(z_f^k, y_f^k)$
5:     $(x^{k,-1}, y^{k,-1}) = (-\mu_x^{-1} z_g^k, y_g^k)$
6:     $x^{k,0} = \mathrm{prox}_{\gamma_x \lambda r(\cdot)}(x^{k,-1} - \gamma_x \lambda a_x^k(x^{k,-1}, y^{k,-1}))$
7:     $y^{k,0} = \mathrm{prox}_{\gamma_y \lambda g(\cdot)}(y^{k,-1} - \gamma_y \lambda a_y^k(x^{k,-1}, y^{k,-1}))$
8:     $b_x^{k,0} = \frac{1}{\gamma_x \lambda}(x^{k,-1} - \gamma_x \lambda a_x^k(x^{k,-1}, y^{k,-1}) - x^{k,0})$
9:     $b_y^{k,0} = \frac{1}{\gamma_y \lambda}(y^{k,-1} - \gamma_y \lambda a_y^k(x^{k,-1}, y^{k,-1}) - y^{k,0})$
10:     $t = 0$
11:     **while** condition (31) is not satisfied **do**
12:         $x^{k,t+1/2} = x^{k,t} + \beta_t(x^{k,0} - x^{k,t}) - \gamma_x \lambda(a_x^k(x^{k,t}, y^{k,t}) + b_x^{k,t})$
13:         $y^{k,t+1/2} = y^{k,t} + \beta_t(y^{k,0} - y^{k,t}) - \gamma_y \lambda(a_y^k(x^{k,t}, y^{k,t}) + b_y^{k,t})$
14:         $x^{k,t+1} = \mathrm{prox}_{\gamma_x \lambda r(\cdot)}(x^{k,t} + \beta_t(x^{k,0} - x^{k,t}) - \gamma_x \lambda a_x^k(x^{k,t+1/2}, y^{k,t+1/2}))$
15:         $y^{k,t+1} = \mathrm{prox}_{\gamma_y \lambda g(\cdot)}(y^{k,t} + \beta_t(y^{k,0} - y^{k,t}) - \gamma_y \lambda a_y^k(x^{k,t+1/2}, y^{k,t+1/2}))$
16:         $b_x^{k,t+1} = \frac{1}{\gamma_x \lambda}(x^{k,t} + \beta_t(x^{k,0} - x^{k,t}) - \gamma_x \lambda a_x^k(x^{k,t+1/2}, y^{k,t+1/2}) - x^{k,t+1})$
17:         $b_y^{k,t+1} = \frac{1}{\gamma_y \lambda}(y^{k,t} + \beta_t(y^{k,0} - y^{k,t}) - \gamma_y \lambda a_y^k(x^{k,t+1/2}, y^{k,t+1/2}) - y^{k,t+1})$
18:         $t = t + 1$
19:     **end while**
20:     $t^k = t$
21:     $(x_f^{k+1}, y_f^{k+1}) = (x^{k,t^k}, y^{k,t^k})$
22:     $(z_f^{k+1}, w_f^{k+1}) = (\nabla_x \hat{F}(x_f^{k+1}, y_f^{k+1}) + b_x^{k,t^k}, -\nabla_y \hat{F}(x_f^{k+1}, y_f^{k+1}) + b_y^{k,t^k})$
23:     $z^{k+1} = z^k + \eta_z \mu_x^{-1}(z_f^{k+1} - z^k) - \eta_z(x_f^{k+1} + \mu_x^{-1} z_f^{k+1})$
24:     $y^{k+1} = y^k + \eta_y \mu_y(y_f^{k+1} - y^k) - \eta_y(w_f^{k+1} + \mu_y y_f^{k+1})$
25: **end for**
26: **output:** $(-\mu_x^{-1} z^K, y^K)$
---

We also set the initial iterates to $x^{k,-1} = -\mu_x^{-1} z_g^k$ and $y^{k,-1} = y_g^k$ on line 5 of Algorithm 4, and use the output of the inner while-loop to compute vectors $(x_f^{k+1}, y_f^{k+1}, z_f^{k+1}, w_f^{k+1})$ on lines 21 and 22 of Algorithm 4. Now, if we define parameters $\gamma_x, \gamma_y$ in the following way:

$$\gamma_x = 8\mu_x^{-1}, \qquad \gamma_y = \theta_y, \tag{32}$$

then condition (31) on line 11 of Algorithm 4 becomes equivalent to condition (18) on line 5 of Algorithm 2. Hence, vectors $(x_f^{k+1}, y_f^{k+1}, z_f^{k+1}, w_f^{k+1})$ computed on lines 21 and 22 of Algorithm 4 satisfy condition (18), which implies that Algorithm 4 is a special case of Algorithm 2.

## 6.2 Complexity of the Algorithm

It remains to establish the gradient evaluation complexity of Algorithm 4. First, we need to estimate the number of iterations performed by the inner while-loop of Algorithm 4, which is equal to $t^k$ defined on line 20 of Algorithm 4. Recall that the inner while-loop was constructed out of the lines of Algorithm 3. Hence, we can use Theorem 2 to provide an upper bound on $t^k$. This is done by the following lemma.

**Lemma 4.** *Assume the following choice of the parameters of Algorithm 4: stepsize $\lambda$ is defined by (25), parameter $M$ is defined by (30), sequence $\{\beta_t\}_{t=0}^{\infty}$ is defined by (24), parameters $\gamma_x$ and $\gamma_y$ are defined by (32). Then, $t^k \leq T$, where $T$ is given as*

$$T = \lceil 48\sqrt{2} \max\{8L/\mu_x, 1 + \theta_y L\} \rceil - 1. \tag{33}$$

Now, we are ready to provide the final gradient complexity of Algorithm 4. It is done by the following theorem.

**Theorem 3.** *Let parameters of Algorithm 4 be defined as follows:* $\alpha = \min\left\{1, \sqrt{\theta_y \mu_y}\right\}$, $\theta_y = 8\mu_x^{-1}$, $\lambda = \left(2\sqrt{5}(1 + 8L/\mu_x)\right)^{-1}$, *stepsizes $\eta_z$ and $\eta_y$ are defined by* (20)*, parameters $\gamma_x$ and $\gamma_y$ are defined by* (32)*, parameters $\{\beta_t\}_{t=0}^{\infty}$ are defined by* (24)*. Then, to find an $\epsilon$-accurate solution of problem* (1)*, Algorithm 4 requires the following number of gradient evaluations:*

$$\mathcal{O}\left(\max\left\{\frac{L}{\mu_x}, \frac{L}{\sqrt{\mu_x \mu_y}}\right\} \log \frac{1}{\epsilon}\right). \tag{34}$$

**Corollary 1.** *Without loss of generality we can assume $\mu_x \geq \mu_y$, otherwise we just swap variables $x$ and $y$ in problem* (1)*. Hence, Algorithm 4 has the following gradient evaluation complexity:*

$$\mathcal{O}\left(\frac{L}{\sqrt{\mu_x \mu_y}} \log \frac{1}{\epsilon}\right). \tag{35}$$

## Acknowledgements

The work of A. Gasnikov was supported by a grant for research centers in the field of artificial intelligence, provided by the Analytical Center for the Government of the Russian Federation in accordance with the subsidy agreement (agreement identifier 000000D730321P5Q0002) and the agreement with the Ivannikov Institute for System Programming of the Russian Academy of Sciences dated November 2, 2021 No. 70-2021-00142.

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
