# Appendix

## A  Further Discussion

**Different smoothness constants with respect to variables $x$ and $y$.**  In our work, we use the smoothness assumption (Assumption 3) with a single smoothness constant $L > 0$. However, there is another common minimax optimization setting in which different smoothness constants are assumed for variables $x$ and $y$. In particular, it is often assumed that there exist constants $L_x, L_y, L_{xy} > 0$, such that for all $x_1, x_2, x \in \mathbb{R}_x^d$ and for all $y_1, y_2, y \in \mathbb{R}^{d_y}$, the following inequalities hold:

$$
\begin{aligned}
\|\nabla_x F(x_1, y) - \nabla_x F(x_2, y)\| &\leq L_x \|x_1 - x_2\|, \\
\|\nabla_x F(x, y_1) - \nabla_x F(x, y_2)\| &\leq L_{xy} \|y_1 - y_2\|, \\
\|\nabla_y F(x_1, y) - \nabla_y F(x_2, y)\| &\leq L_{xy} \|x_1 - x_2\|, \\
\|\nabla_y F(x, y_1) - \nabla_y F(x, y_2)\| &\leq L_y \|y_1 - y_2\|.
\end{aligned}
\tag{36}
$$

Assumption 3 follows from these inequalities if $L = 2 \max\{L_x, L_y, L_{xy}\}$. Hence, the complexity result (35) still holds with this choice of $L$. However, one can obtain the following improved complexity of Algorithm 4 by rescaling variables $x$ and $y$ in problem (1):

$$
\mathcal{O}\left( \frac{\sqrt{L_x L_y} + L_{xy}}{\sqrt{\mu_x \mu_y}} \log \frac{1}{\epsilon} \right).
\tag{37}
$$

Unfortunately, we cannot prove that this complexity of Algorithm 4 is optimal in this setting. In particular, the complexity result (37) does not match the lower bound of Zhang et al. (2021)

$$
\Omega\left( \left( \sqrt{\frac{L_x}{\mu_x}} + \sqrt{\frac{L_y}{\mu_y}} + \frac{L_{xy}}{\sqrt{\mu_x \mu_y}} \right) \log \frac{1}{\epsilon} \right).
\tag{38}
$$

However, at this moment, it is unclear whether the lower bound of Zhang et al. (2021) is tight because this lower bound belongs to the class of minimax optimization problems with bilinear coupling. It is an open question whether it is possible to find a better lower bound outside of this class.

**Minimax optimization problems with bilinear coupling.**  As mentioned in Section 1.1, the lower complexity bound (38) of Zhang et al. (2021) is achieved by Kovalev et al. (2021); Thekumparampil et al. (2022); Jin et al. (2022) for the class of minimax problems with bilinear coupling of the form

$$
\min_{x \in \mathbb{R}^{d_x}} \max_{y \in \mathbb{R}^{d_y}} p(x) + x^\top \mathbf{A} y - q(y).
\tag{39}
$$

However, the algorithms and techniques used in these works are not applicable to the general minimax problem (1). In particular, these algorithms are based on the forward-backward algorithm for solving a monotone inclusion problem $0 = G(x^*, y^*) + H(x^*, y^*)$, where monotone operators $G(x, y)$ and $H(x, y)$ are defined as $G(x, y) = (\nabla p(x), \nabla q(y)), H(x, y) = (\mathbf{A}x, -\mathbf{A}^\top y)$. The main observation here is that operator $G(x, y)$ is potential, i.e., it is equal to the gradient of a smooth convex function $(x, y) \mapsto p(x) + q(y)$. Hence, it is possible to apply Nesterov acceleration.

Unfortunately, these ideas cannot be applied to solving general minimax optimization problem (1) because the potential functions $p(x)$ and $q(y)$ are, in some sense, "hidden" inside the function $F(x, y)$. It is an important open question whether the lower complexity bound (38) can be achieved in the case of general smooth strongly-convex-strongly-concave minimax optimization problems.

# B   Proof of Lemma 1

From the definition of $G(z, y)$ it follows that

$$G(z, y) \geq \langle z, x \rangle - r(x) - \hat{F}(x, y) + g(y). \tag{40}$$

Using $z - \nabla_x \hat{F}(x, y) \in \partial r(x)$, for arbitrary $\bar{x} \in \mathbb{R}^{d_x}$ we get

$$r(\bar{x}) \geq r(x) + \langle z - \nabla_x \hat{F}(x, y), \bar{x} - x \rangle$$
$$\geq r(x) + \langle z, \bar{x} - x \rangle + \hat{F}(x, y) - \hat{F}(\bar{x}, y),$$

where we used the convexity of $\hat{F}(x, y)$ in $x$ in the last inequality. After rearranging we get

$$\langle z, x \rangle - r(x) - \hat{F}(x, y) \geq \langle z, \bar{x} \rangle - r(\bar{x}) - \hat{F}(\bar{x}, y).$$

Now, we add $g(y)$ to both sides of the inequality and take supremum over $\bar{x} \in \mathbb{R}^{d_x}$. This gives us

$$\langle z, x \rangle - r(x) - \hat{F}(x, y) + g(y) \geq \sup_{\bar{x} \in \mathbb{R}^{d_x}} \left[ \langle z, \bar{x} \rangle - r(\bar{x}) - \hat{F}(\bar{x}, y) + g(y) \right] = G(z, y),$$

which together with (40) implies $G(z, y) = \langle z, x \rangle - r(x) - \hat{F}(x, y) + g(y)$.

Next, we use $w + \nabla_y \hat{F}(x, y) \in \partial g(y)$, which for arbitrary $\bar{y} \in \mathbb{R}^{d_y}$ implies

$$g(\bar{y}) \geq g(y) + \langle w + \nabla_y \hat{F}(x, y), \bar{y} - y \rangle$$
$$\geq g(y) + \langle w, \bar{y} - y \rangle + \hat{F}(x, \bar{y}) - \hat{F}(x, y),$$

where we used the concavity of $\hat{F}(x, y)$ in $y$ in the last inequality. After rearranging we get

$$g(\bar{y}) - \hat{F}(x, \bar{y}) \geq g(y) - \hat{F}(x, y) + \langle w, \bar{y} - y \rangle.$$

Now, we choose arbitrary $\bar{z} \in \mathbb{R}^{d_x}$ and add $\langle x, \bar{z} \rangle - r(x)$ to both sides of the inequality, which implies

$$g(\bar{y}) - \hat{F}(x, \bar{y}) + \langle x, \bar{z} \rangle - r(x) \geq g(y) - \hat{F}(x, y) + \langle w, \bar{y} - y \rangle + \langle x, \bar{z} \rangle - r(x)$$
$$= G(z, y) + \langle x, \bar{z} - z \rangle + \langle w, \bar{y} - y \rangle,$$

where we used $G(z, y) = \langle z, x \rangle - r(x) - \hat{F}(x, y) + g(y)$. Further, using (40) we get

$$G(z, y) + \langle x, \bar{z} - z \rangle + \langle w, \bar{y} - y \rangle \leq G(\bar{z}, \bar{y}),$$

which holds for arbitrary $\bar{z} \in \mathbb{R}^{d_x}, \bar{y} \in \mathbb{R}^{d_y}$. Hence, $(x, w) \in \partial G(z, y)$ by the definition of the subdifferential of a convex function. □

# C   Proof of Lemma 2

From the optimality conditions (5) and the definition of $\hat{F}(x, y)$, it follows that

$$-\mu_x x^* - \nabla_x \hat{F}(x^*, y^*) \in \partial r(x^*),$$
$$-\mu_y y^* + \nabla_y \hat{F}(x^*, y^*) \in \partial g(y^*).$$

Using (11) we get

$$z^* - \nabla_x \hat{F}(x^*, y^*) \in \partial r(x^*),$$
$$-\mu_y y^* + \nabla_y \hat{F}(x^*, y^*) \in \partial g(y^*).$$

Using Lemma 1 we get

$$(x^*, -\mu_y y^*) \in \partial G(z^*, y^*),$$

which together with (11) implies

$$(-\mu_x^{-1} z^*, -\mu_y y^*) \in \partial G(z^*, y^*).$$

The latter condition implies $0 \in \partial P(z^*, y^*)$. Hence, $(z^*, y^*)$ is indeed a solution of problem (10). The uniqueness of this solution is implied by the strong convexity of the function $P(z, y)$. □

## D   Proof of Theorem 1

We start with proving two technical lemmas.

**Lemma 5.** *Under conditions of Theorem 1 the following inequality holds:*

$$\frac{1}{\eta_z}\|z^{k+1}-z^*\|^2 \leq \left(\frac{1}{\eta_z}-\mu_x^{-1}\right)\|z^k-z^*\|^2 + \mu_x^{-1}\|z_f^{k+1}-z^*\|^2$$

$$+ \frac{2}{\alpha}\langle x_f^{k+1}+\mu_x^{-1}z_f^{k+1},(1-\alpha)z_f^k+\alpha z^*-z_f^{k+1}\rangle \qquad (41)$$

$$+ \frac{1}{\alpha}\left(\frac{8}{\mu_x}\|z_f^{k+1}\|^2 + \frac{\mu_x}{2}\|(x_f^{k+1}-\mu_x^{-1}z_g^k)\|^2 - \frac{\mu_x}{8}\|x_f^{k+1}+\mu_x^{-1}z_g^k\|^2\right).$$

*Proof.* Using line 6 of Algorithm 2 we get

$$\frac{1}{\eta_z}\|z^{k+1}-z^*\|^2 = \frac{1}{\eta_z}\|z^k-z^*\|^2 + \frac{2}{\eta_z}\langle z^{k+1}-z^k, z^k-z^*\rangle + \frac{1}{\eta_z}\|z^{k+1}-z^k\|^2$$

$$= \frac{1}{\eta_z}\|z^k-z^*\|^2 + \eta_z\|\mu_x^{-1}(z_f^{k+1}-z^k)-(x_f^{k+1}+\mu_x^{-1}z_f^{k+1})\|^2$$

$$+ 2\mu_x^{-1}\langle z_f^{k+1}-z^k, z^k-z^*\rangle - 2\langle x_f^{k+1}+\mu_x^{-1}z_f^{k+1}, z^k-z^*\rangle.$$

Using the parallelogram rule we get

$$\frac{1}{\eta_z}\|z^{k+1}-z^*\|^2 = \frac{1}{\eta_z}\|z^k-z^*\|^2 + \eta_z\|\mu_x^{-1}(z_f^{k+1}-z^k)-(x_f^{k+1}+\mu_x^{-1}z_f^{k+1})\|^2$$

$$+ \mu_x^{-1}\|z_f^{k+1}-z^*\|^2 - \mu_x^{-1}\|z^k-z^*\|^2 - \mu_x^{-1}\|z_f^{k+1}-z^k\|^2$$

$$- 2\langle x_f^{k+1}+\mu_x^{-1}z_f^{k+1}, z^k-z^*\rangle.$$

Using the inequality $\|a+b\|^2 \leq 2\|a\|^2 + 2\|b\|^2$ we get

$$\frac{1}{\eta_z}\|z^{k+1}-z^*\|^2 \leq \frac{1}{\eta_z}\|z^k-z^*\|^2 + 2\eta_z\|x_f^{k+1}+\mu_x^{-1}z_f^{k+1}\|^2 + 2\eta_z\mu_x^{-2}\|z_f^{k+1}-z^k\|^2$$

$$+ \mu_x^{-1}\|z_f^{k+1}-z^*\|^2 - \mu_x^{-1}\|z^k-z^*\|^2 - \mu_x^{-1}\|z_f^{k+1}-z^k\|^2$$

$$- 2\langle x_f^{k+1}+\mu_x^{-1}z_f^{k+1}, z^k-z^*\rangle.$$

Using the definition of $\eta_z$ we get

$$\frac{1}{\eta_z}\|z^{k+1}-z^*\|^2 \leq \frac{1}{\eta_z}\|z^k-z^*\|^2 + \mu_x\|x_f^{k+1}+\mu_x^{-1}z_f^{k+1}\|^2 + \mu_x^{-1}\|z_f^{k+1}-z^k\|^2$$

$$+ \mu_x^{-1}\|z_f^{k+1}-z^*\|^2 - \mu_x^{-1}\|z^k-z^*\|^2 - \mu_x^{-1}\|z_f^{k+1}-z^k\|^2$$

$$- 2\langle x_f^{k+1}+\mu_x^{-1}z_f^{k+1}, z^k-z^*\rangle$$

$$= \left(\frac{1}{\eta_z}-\mu_x^{-1}\right)\|z^k-z^*\|^2 + \mu_x\|x_f^{k+1}+\mu_x^{-1}z_f^{k+1}\|^2$$

$$+ \mu_x^{-1}\|z_f^{k+1}-z^*\|^2 - 2\langle x_f^{k+1}+\mu_x^{-1}z_f^{k+1}, z^k-z^*\rangle.$$

From line 4 of Algorithm 2 we get $z^k = \alpha^{-1}z_g^k - (1-\alpha)\alpha^{-1}z_f^k$ which implies

$$\frac{1}{\eta_z}\|z^{k+1}-z^*\|^2 \leq \left(\frac{1}{\eta_z}-\mu_x^{-1}\right)\|z^k-z^*\|^2 + \mu_x\|x_f^{k+1}+\mu_x^{-1}z_f^{k+1}\|^2$$

$$+ \mu_x^{-1}\|z_f^{k+1}-z^*\|^2 - 2\langle x_f^{k+1}+\mu_x^{-1}z_f^{k+1}, \alpha^{-1}z_g^k-(1-\alpha)\alpha^{-1}z_f^k-z^*\rangle$$

$$= \left(\frac{1}{\eta_z}-\mu_x^{-1}\right)\|z^k-z^*\|^2 + \mu_x\|x_f^{k+1}+\mu_x^{-1}z_f^{k+1}\|^2 + \mu_x^{-1}\|z_f^{k+1}-z^*\|^2$$

$$+ \frac{2}{\alpha}\langle x_f^{k+1}+\mu_x^{-1}z_f^{k+1},(1-\alpha)z_f^k+\alpha z^*-z_f^{k+1}\rangle$$

$$+ \frac{2}{\alpha}\langle x_f^{k+1}+\mu_x^{-1}z_f^{k+1}, z_f^{k+1}-z_g^k\rangle$$

$$= \left( \frac{1}{\eta_z} - \mu_x^{-1} \right) \|z^k - z^*\|^2 + \mu_x \|x_f^{k+1} + \mu_x^{-1} z_f^{k+1}\|^2 + \mu_x^{-1} \|z_f^{k+1} - z^*\|^2$$

$$+ \frac{2}{\alpha} \langle x_f^{k+1} + \mu_x^{-1} z_f^{k+1}, (1-\alpha) z_f^k + \alpha z^* - z_f^{k+1} \rangle$$

$$+ \frac{2\mu_x}{\alpha} \langle x_f^{k+1} + \mu_x^{-1} z_f^{k+1}, \mu_x^{-1} (z_f^{k+1} - z_g^k) \rangle.$$

Using the parallelogram rule we get

$$\frac{1}{\eta_z} \|z^{k+1} - z^*\|^2 \le \left( \frac{1}{\eta_z} - \mu_x^{-1} \right) \|z^k - z^*\|^2 + \mu_x \|x_f^{k+1} + \mu_x^{-1} z_f^{k+1}\|^2 + \mu_x^{-1} \|z_f^{k+1} - z^*\|^2$$

$$+ \frac{2}{\alpha} \langle x_f^{k+1} + \mu_x^{-1} z_f^{k+1}, (1-\alpha) z_f^k + \alpha z^* - z_f^{k+1} \rangle$$

$$+ \frac{\mu_x}{\alpha} \|x_f^{k+1} + 2\mu_x^{-1} z_f^{k+1} - \mu_x^{-1} z_g^k\|^2 - \frac{\mu_x^{-1}}{\alpha} \|z_f^{k+1} - z_g^k\|^2 - \frac{\mu_x}{\alpha} \|x_f^{k+1} + \mu_x^{-1} z_g^k\|^2$$

$$= \left( \frac{1}{\eta_z} - \mu_x^{-1} \right) \|z^k - z^*\|^2 + \mu_x (1 - \alpha^{-1}) \|x_f^{k+1} + \mu_x^{-1} z_f^{k+1}\|^2 + \mu_x^{-1} \|z_f^{k+1} - z^*\|^2$$

$$+ \frac{2}{\alpha} \langle x_f^{k+1} + \mu_x^{-1} z_f^{k+1}, (1-\alpha) z_f^k + \alpha z^* - z_f^{k+1} \rangle$$

$$+ \frac{4\mu_x^{-1}}{\alpha} \|z_f^{k+1} + \frac{\mu_x}{2} (x_f^{k+1} - \mu_x^{-1} z_g^k)\|^2 - \frac{\mu_x^{-1}}{\alpha} \|z_f^{k+1} - z_g^k\|^2.$$

Using the fact that $\alpha^{-1} \ge 1$ we get

$$\frac{1}{\eta_z} \|z^{k+1} - z^*\|^2 \le \left( \frac{1}{\eta_z} - \mu_x^{-1} \right) \|z^k - z^*\|^2 + \mu_x^{-1} \|z_f^{k+1} - z^*\|^2$$

$$+ \frac{2}{\alpha} \langle x_f^{k+1} + \mu_x^{-1} z_f^{k+1}, (1-\alpha) z_f^k + \alpha z^* - z_f^{k+1} \rangle$$

$$+ \frac{4\mu_x^{-1}}{\alpha} \|z_f^{k+1} + \frac{\mu_x}{2} (x_f^{k+1} - \mu_x^{-1} z_g^k)\|^2 - \frac{\mu_x^{-1}}{\alpha} \|z_f^{k+1} - z_g^k\|^2$$

$$= \left( \frac{1}{\eta_z} - \mu_x^{-1} \right) \|z^k - z^*\|^2 + \mu_x^{-1} \|z_f^{k+1} - z^*\|^2$$

$$+ \frac{2}{\alpha} \langle x_f^{k+1} + \mu_x^{-1} z_f^{k+1}, (1-\alpha) z_f^k + \alpha z^* - z_f^{k+1} \rangle$$

$$+ \frac{4\mu_x^{-1}}{\alpha} \|z_f^{k+1} + \frac{\mu_x}{2} (x_f^{k+1} - \mu_x^{-1} z_g^k)\|^2$$

$$- \frac{\mu_x^{-1}}{\alpha} \|z_f^{k+1} + \frac{\mu_x}{2} (x_f^{k+1} - \mu_x^{-1} z_g^k) - \frac{\mu_x}{2} (x_f^{k+1} - \mu_x^{-1} z_g^k) - z_g^k\|^2.$$

Using the inequality $-\|a + b\|^2 \le \|b\|^2 - \frac{1}{2} \|a\|^2$ we get

$$\frac{1}{\eta_z} \|z^{k+1} - z^*\|^2 \le \left( \frac{1}{\eta_z} - \mu_x^{-1} \right) \|z^k - z^*\|^2 + \mu_x^{-1} \|z_f^{k+1} - z^*\|^2$$

$$+ \frac{2}{\alpha} \langle x_f^{k+1} + \mu_x^{-1} z_f^{k+1}, (1-\alpha) z_f^k + \alpha z^* - z_f^{k+1} \rangle$$

$$+ \frac{4\mu_x^{-1}}{\alpha} \|z_f^{k+1} + \frac{\mu_x}{2} (x_f^{k+1} - \mu_x^{-1} z_g^k)\|^2$$

$$+ \frac{\mu_x^{-1}}{\alpha} \|z_f^{k+1} + \frac{\mu_x}{2} (x_f^{k+1} - \mu_x^{-1} z_g^k)\|^2 - \frac{\mu_x^{-1}}{2\alpha} \|\frac{\mu_x}{2} (x_f^{k+1} - \mu_x^{-1} z_g^k) + z_g^k\|^2$$

$$= \left( \frac{1}{\eta_z} - \mu_x^{-1} \right) \|z^k - z^*\|^2 + \mu_x^{-1} \|z_f^{k+1} - z^*\|^2$$

$$+ \frac{2}{\alpha} \langle x_f^{k+1} + \mu_x^{-1} z_f^{k+1}, (1-\alpha) z_f^k + \alpha z^* - z_f^{k+1} \rangle$$

$$+ \frac{5\mu_x^{-1}}{\alpha} \|z_f^{k+1} + \frac{\mu_x}{2} (x_f^{k+1} - \mu_x^{-1} z_g^k)\|^2 - \frac{\mu_x}{8\alpha} \|x_f^{k+1} + \mu_x^{-1} z_g^k\|^2.$$

Using the inequality $5 \le 8$ concludes the proof $\qquad\square$

**Lemma 6.** *Under conditions of Theorem 1 the following inequality holds:*

$$\frac{1}{\eta_y}\|y^{k+1} - y^*\|^2 \le \left(\frac{1}{\eta_y} - \mu_y\right)\|y^k - y^*\|^2 + \mu_y\|y_f^{k+1} - y^*\|^2$$

$$+ \frac{2}{\alpha}\langle w_f^{k+1} + \mu_y y_f^{k+1}, (1-\alpha)y_f^k + \alpha y^* - y_f^{k+1}\rangle \qquad (42)$$

$$+ \frac{1}{\alpha}\left(\theta_y\|w_f^{k+1} + \mu_y y_f^{k+1} + \theta_y^{-1}(y_f^{k+1} - y_g^k)\|^2 - \theta_y^{-1}\|y_f^{k+1} - y_g^k\|^2\right).$$

*Proof.* Using line 7 of Algorithm 2 we get

$$\frac{1}{\eta_y}\|y^{k+1} - y^*\|^2 = \frac{1}{\eta_y}\|y^k - y^*\|^2 + \frac{2}{\eta_y}\langle y^{k+1} - y^k, y^k - y^*\rangle + \frac{1}{\eta_y}\|y^{k+1} - y^*\|^2$$

$$= \frac{1}{\eta_y}\|y^k - y^*\|^2 + \eta_y\|\mu_y(y_f^{k+1} - y^k) - (w_f^{k+1} + \mu_y y_f^{k+1})\|^2$$

$$+ 2\mu_y\langle y_f^{k+1} - y^k, y^k - y^*\rangle - 2\langle w_f^{k+1} + \mu_y y_f^{k+1}, y^k - y^*\rangle.$$

Using the parallelogram rule we get

$$\frac{1}{\eta_y}\|y^{k+1} - y^*\|^2 = \frac{1}{\eta_y}\|y^k - y^*\|^2 + \eta_y\|\mu_y(y_f^{k+1} - y^k) - (w_f^{k+1} + \mu_y y_f^{k+1})\|^2$$

$$\mu_y\|y_f^{k+1} - y^*\|^2 - \mu_y\|y^k - y^*\|^2 - \mu_y\|y_f^{k+1} - y^k\|^2$$

$$- 2\langle w_f^{k+1} + \mu_y y_f^{k+1}, y^k - y^*\rangle.$$

Using the inequality $\|a + b\|^2 \le 2\|a\|^2 + 2\|b\|^2$ we get

$$\frac{1}{\eta_y}\|y^{k+1} - y^*\|^2 = \frac{1}{\eta_y}\|y^k - y^*\|^2 + 2\eta_y\|w_f^{k+1} + \mu_y y_f^{k+1}\|^2 + 2\eta_y\mu_y^2\|y_f^{k+1} - y^k\|^2$$

$$+ \mu_y\|y_f^{k+1} - y^*\|^2 - \mu_y\|y^k - y^*\|^2 - \mu_y\|y_f^{k+1} - y^k\|^2$$

$$- 2\langle w_f^{k+1} + \mu_y y_f^{k+1}, y^k - y^*\rangle.$$

Using the definition of $\eta_y$ we get

$$\frac{1}{\eta_y}\|y^{k+1} - y^*\|^2 \le \frac{1}{\eta_y}\|y^k - y^*\|^2 + 2\eta_y\|w_f^{k+1} + \mu_y y_f^{k+1}\|^2 + \mu_y\|y_f^{k+1} - y^k\|^2$$

$$+ \mu_y\|y_f^{k+1} - y^*\|^2 - \mu_y\|y^k - y^*\|^2 - \mu_y\|y_f^{k+1} - y^k\|^2$$

$$- 2\langle w_f^{k+1} + \mu_y y_f^{k+1}, y^k - y^*\rangle$$

$$= \left(\frac{1}{\eta_y} - \mu_y\right)\|y^k - y^*\|^2 + 2\eta_y\|w_f^{k+1} + \mu_y y_f^{k+1}\|^2$$

$$+ \mu_y\|y_f^{k+1} - y^*\|^2 - 2\langle w_f^{k+1} + \mu_y y_f^{k+1}, y^k - y^*\rangle.$$

From line 4 of Algorithm 2 we get $y^k = \alpha^{-1}y_g^k - (1-\alpha)\alpha^{-1}y_f^k$, which implies

$$\frac{1}{\eta_y}\|y^{k+1} - y^*\|^2 \le \left(\frac{1}{\eta_y} - \mu_y\right)\|y^k - y^*\|^2 + 2\eta_y\|w_f^{k+1} + \mu_y y_f^{k+1}\|^2$$

$$+ \mu_y\|y_f^{k+1} - y^*\|^2 - 2\langle w_f^{k+1} + \mu_y y_f^{k+1}, \alpha^{-1}y_g^k - (1-\alpha)\alpha^{-1}y_f^k - y^*\rangle$$

$$= \left(\frac{1}{\eta_y} - \mu_y\right)\|y^k - y^*\|^2 + 2\eta_y\|w_f^{k+1} + \mu_y y_f^{k+1}\|^2 + \mu_y\|y_f^{k+1} - y^*\|^2$$

$$+ \frac{2}{\alpha}\langle w_f^{k+1} + \mu_y y_f^{k+1}, (1-\alpha)y_f^k + \alpha y^* - y_f^{k+1}\rangle$$

$$+ \frac{2}{\alpha}\langle w_f^{k+1} + \mu_y y_f^{k+1}, y_f^{k+1} - y_g^k\rangle$$

$$= \left(\frac{1}{\eta_y} - \mu_y\right)\|y^k - y^*\|^2 + 2\eta_y\|w_f^{k+1} + \mu_y y_f^{k+1}\|^2 + \mu_y\|y_f^{k+1} - y^*\|^2$$

$$+ \frac{2}{\alpha} \langle w_f^{k+1} + \mu_y y_f^{k+1}, (1-\alpha)y_f^k + \alpha y^* - y_f^{k+1} \rangle$$

$$+ \frac{2\theta_y}{\alpha} \langle w_f^{k+1} + \mu_y y_f^{k+1}, \theta_y^{-1}(y_f^{k+1} - y_g^k) \rangle$$

Using the parallelogram rule we get

$$\frac{1}{\eta_y}\|y^{k+1} - y^*\|^2 \leq \left(\frac{1}{\eta_y} - \mu_y\right)\|y^k - y^*\|^2 + 2\eta_y\|w_f^{k+1} + \mu_y y_f^{k+1}\|^2 + \mu_y\|y_f^{k+1} - y^*\|^2$$

$$+ \frac{2}{\alpha}\langle w_f^{k+1} + \mu_y y_f^{k+1}, (1-\alpha)y_f^k + \alpha y^* - y_f^{k+1}\rangle$$

$$+ \frac{\theta_y}{\alpha}\|w_f^{k+1} + \mu_y y_f^{k+1} + \theta_y^{-1}(y_f^{k+1} - y_g^k)\|^2$$

$$- \frac{\theta_y}{\alpha}\|w_f^{k+1} + \mu_y y_f^{k+1}\|^2 - \frac{\theta_y^{-1}}{\alpha}\|y_f^{k+1} - y_g^k\|^2$$

$$= \left(\frac{1}{\eta_y} - \mu_y\right)\|y^k - y^*\|^2 + (2\eta_y - \alpha^{-1}\theta_y)\|w_f^{k+1} + \mu_y y_f^{k+1}\|^2 + \mu_y\|y_f^{k+1} - y^*\|^2$$

$$+ \frac{2}{\alpha}\langle w_f^{k+1} + \mu_y y_f^{k+1}, (1-\alpha)y_f^k + \alpha y^* - y_f^{k+1}\rangle$$

$$+ \frac{\theta_y}{\alpha}\|w_f^{k+1} + \mu_y y_f^{k+1} + \theta_y^{-1}(y_f^{k+1} - y_g^k)\|^2 - \frac{\theta_y^{-1}}{\alpha}\|y_f^{k+1} - y_g^k\|^2.$$

From the definition of $\eta_y$ it follows that $2\eta_y \leq \alpha^{-1}\theta_y$. Hence,

$$\frac{1}{\eta_y}\|y^{k+1} - y^*\|^2 \leq \left(\frac{1}{\eta_y} - \mu_y\right)\|y^k - y^*\|^2 + \mu_y\|y_f^{k+1} - y^*\|^2$$

$$+ \frac{2}{\alpha}\langle w_f^{k+1} + \mu_y y_f^{k+1}, (1-\alpha)y_f^k + \alpha y^* - y_f^{k+1}\rangle$$

$$+ \frac{\theta_y}{\alpha}\|w_f^{k+1} + \mu_y y_f^{k+1} + \theta_y^{-1}(y_f^{k+1} - y_g^k)\|^2 - \frac{\theta_y^{-1}}{\alpha}\|y_f^{k+1} - y_g^k\|^2.$$

$$\square$$

**Lemma 7.** *Under conditions of Theorem 1, let $\mathcal{L}^k$ be the following Lyapunov function*

$$\mathcal{L}^k = \frac{1}{\eta_z}\|z^k - z^*\|^2 + \frac{1}{\eta_y}\|y^k - y^*\|^2 + \frac{2}{\alpha}\left(P(z_f^k, y_f^k) - P(z^*, y^*)\right). \tag{43}$$

*Then, the following inequality holds*

$$\mathcal{L}^{k+1} \leq \left(1 - \max\left\{\frac{2}{\alpha}, \frac{2\alpha}{\theta_y \mu_y}\right\}^{-1}\right)\mathcal{L}^k. \tag{44}$$

*Proof.* We start with combining Lemmas 5 and 6 and get

$$\text{(DISTANCE)} \leq \left(\frac{1}{\eta_z} - \mu_x^{-1}\right)\|z^k - z^*\|^2 + \mu_x^{-1}\|z_f^{k+1} - z^*\|^2$$

$$+ \frac{2}{\alpha}\langle x_f^{k+1} + \mu_x^{-1}z_f^{k+1}, (1-\alpha)z_f^k + \alpha z^* - z_f^{k+1}\rangle$$

$$+ \frac{1}{\alpha}\left(\frac{8}{\mu_x}\|z_f^{k+1} + \frac{\mu_x}{2}(x_f^{k+1} - \mu_x^{-1}z_g^k)\|^2 - \frac{\mu_x}{8}\|x_f^{k+1} + \mu_x^{-1}z_g^k\|^2\right)$$

$$+ \left(\frac{1}{\eta_y} - \mu_y\right)\|y^k - y^*\|^2 + \mu_y\|y_f^{k+1} - y^*\|^2$$

$$+ \frac{2}{\alpha}\langle w_f^{k+1} + \mu_y y_f^{k+1}, (1-\alpha)y_f^k + \alpha y^* - y_f^{k+1}\rangle$$

$$+ \frac{1}{\alpha}\left(\theta_y\|w_f^{k+1} + \mu_y y_f^{k+1} + \theta_y^{-1}(y_f^{k+1} - y_g^k)\|^2 - \theta_y^{-1}\|y_f^{k+1} - y_g^k\|^2\right).$$

where (DISTANCE) is defined as

$$(\text{DISTANCE}) = \frac{1}{\eta_z}\|z^{k+1} - z^*\|^2 + \frac{1}{\eta_y}\|y^{k+1} - y^*\|^2.$$

Using condition (18) we get

$$
\begin{aligned}
(\text{DISTANCE}) \leq{}& \left(\frac{1}{\eta_z} - \mu_x^{-1}\right)\|z^k - z^*\|^2 + \left(\frac{1}{\eta_y} - \mu_y\right)\|y^k - y^*\|^2 \\
&+ \mu_x^{-1}\|z_f^{k+1} - z^*\|^2 + \mu_y\|y_f^{k+1} - y^*\|^2 \\
&+ \frac{2}{\alpha}\langle x_f^{k+1} + \mu_x^{-1}z_f^{k+1}, (1-\alpha)z_f^k + \alpha z^* - z_f^{k+1}\rangle \\
&+ \frac{2}{\alpha}\langle w_f^{k+1} + \mu_y y_f^{k+1}, (1-\alpha)y_f^k + \alpha y^* - y_f^{k+1}\rangle \\
={}& \left(\frac{1}{\eta_z} - \mu_x^{-1}\right)\|z^k - z^*\|^2 + \left(\frac{1}{\eta_y} - \mu_y\right)\|y^k - y^*\|^2 \\
&+ \mu_x^{-1}\|z_f^{k+1} - z^*\|^2 + \mu_y\|y_f^{k+1} - y^*\|^2 \\
&+ \frac{2(1-\alpha)}{\alpha}\left\langle \begin{bmatrix} x_f^{k+1} \\ w_f^{k+1} \end{bmatrix} + \begin{bmatrix} \mu_x^{-1}z_f^{k+1} \\ \mu_y y_f^{k+1} \end{bmatrix}, \begin{bmatrix} z_f^k \\ y_f^k \end{bmatrix} - \begin{bmatrix} z_f^{k+1} \\ y_f^{k+1} \end{bmatrix} \right\rangle \\
&+ 2\left\langle \begin{bmatrix} x_f^{k+1} \\ w_f^{k+1} \end{bmatrix} + \begin{bmatrix} \mu_x^{-1}z_f^{k+1} \\ \mu_y y_f^{k+1} \end{bmatrix}, \begin{bmatrix} z^* \\ y^* \end{bmatrix} - \begin{bmatrix} z_f^{k+1} \\ y_f^{k+1} \end{bmatrix} \right\rangle.
\end{aligned}
$$

Using condition (18) and Lemma 1 we get

$$\begin{bmatrix} x_f^{k+1} \\ w_f^{k+1} \end{bmatrix} \in \partial G(z_f^{k+1}, y_f^{k+1}).$$

Using the definition of function $P(z, y)$ we get

$$\begin{bmatrix} x_f^{k+1} \\ w_f^{k+1} \end{bmatrix} + \begin{bmatrix} \mu_x^{-1}z_f^{k+1} \\ \mu_y y_f^{k+1} \end{bmatrix} \in \partial P(z_f^{k+1}, y_f^{k+1}). \tag{45}$$

Hence, using the strong convexity of Function $P(z, y)$ we get

$$
\begin{aligned}
(\text{DISTANCE}) \leq{}& \left(\frac{1}{\eta_z} - \mu_x^{-1}\right)\|z^k - z^*\|^2 + \left(\frac{1}{\eta_y} - \mu_y\right)\|y^k - y^*\|^2 \\
&+ \mu_x^{-1}\|z_f^{k+1} - z^*\|^2 + \mu_y\|y_f^{k+1} - y^*\|^2 \\
&+ \frac{2(1-\alpha)}{\alpha}\left(P(z_f^k, y_f^k) - P(z_f^{k+1}, y_f^{k+1})\right) \\
&+ 2\left(P(z^*, y^*) - P(z_f^{k+1}, y_f^{k+1}) - \frac{\mu_x^{-1}}{2}\|z_f^{k+1} - z^*\|^2 - \frac{\mu_y}{2}\|y_f^{k+1} - y^*\|^2\right) \\
={}& \left(\frac{1}{\eta_z} - \mu_x^{-1}\right)\|z^k - z^*\|^2 + \left(\frac{1}{\eta_y} - \mu_y\right)\|y^k - y^*\|^2 \\
&+ \frac{2(1-\alpha)}{\alpha}\left(P(z_f^k, y_f^k) - P(z^*, y^*)\right) - \frac{2}{\alpha}\left(P(z_f^{k+1}, y_f^{k+1}) - P(z^*, y^*)\right).
\end{aligned}
$$

After rearranging and using the definition of (DISTANCE) and the definition of $\mathcal{L}^k$ we get

$$
\begin{aligned}
\mathcal{L}^{k+1} \leq{}& \left(\frac{1}{\eta_z} - \mu_x^{-1}\right)\|z^k - z^*\|^2 + \left(\frac{1}{\eta_y} - \mu_y\right)\|y^k - y^*\|^2 \\
&+ \frac{2(1-\alpha)}{\alpha}\left(P(z_f^k, y_f^k) - P(z^*, y^*)\right) \\
\leq{}& \left(1 - \max\left\{2, \frac{1}{\alpha}, \frac{2\alpha}{\theta_y \mu_y}\right\}^{-1}\right)\mathcal{L}^k \\
\leq{}& \left(1 - \max\left\{\frac{2}{\alpha}, \frac{2\alpha}{\theta_y \mu_y}\right\}^{-1}\right)\mathcal{L}^k.
\end{aligned}
$$

$\square$

Now, we are ready to prove Theorem 1.

**Proof of Theorem 1.** After unrolling the recurrence from Lemma 7 we get

$$\frac{1}{\eta_z}\|z^K - z^*\|^2 + \frac{1}{\eta_y}\|y^K - y^*\|^2 \le \left(1 - \max\left\{\frac{2}{\alpha}, \frac{2\alpha}{\theta_y\mu_y}\right\}^{-1}\right)^K \mathcal{L}^0.$$

Using Lemma 2 we get

$$\left(1 - \max\left\{\frac{2}{\alpha}, \frac{2\alpha}{\theta_y\mu_y}\right\}^{-1}\right)^K \mathcal{L}^0 \ge \frac{1}{\eta_z}\|z^K + \mu_x z^*\|^2 + \frac{1}{\eta_y}\|y^K - y^*\|^2$$

$$= \frac{\mu_x^2}{\eta_z}\|\mu_x^{-1}z^K + z^*\|^2 + \frac{1}{\eta_y}\|y^K - y^*\|^2$$

$$\ge \min\left\{\eta_z^{-1}\mu_x^2, \eta_y^{-1}\right\}\left(\|\mu_x^{-1}z^K + z^*\|^2 + \|y^K - y^*\|^2\right).$$

After rearranging we get

$$\|\mu_x^{-1}z^K + z^*\|^2 + \|y^K - y^*\|^2 \le C\left(1 - \max\left\{\frac{2}{\alpha}, \frac{2\alpha}{\theta_y\mu_y}\right\}^{-1}\right)^K,$$

where $C$ is given as

$$C = \max\{\eta_z\mu_x^{-2}, \eta_y\}\mathcal{L}^0.$$

Hence, $(\|\mu_x^{-1}z^K + z^*\|^2 + \|y^K - y^*\|^2) \le \epsilon$ if the number of iterations $K$ satisfies

$$K \ge \max\left\{\frac{2}{\alpha}, \frac{2\alpha}{\theta_y\mu_y}\right\}\log\frac{C}{\epsilon}.$$

This concludes the proof.

$\square$

# E    Proof of Theorem 2

We start the proof with three technical lemmas.

**Lemma 8.** *Under assumptions of Theorem 2 the following inequality holds:*

$$96M^2\|u^0 - u^*\|^2 + 12\|a^0 + b^0\|^2 \le 288M^2\|u^{-1} - u^*\|^2. \tag{46}$$

*Proof.* From Line 3 of Algorithm 3 it follows that

$$u^0 = \mathrm{J}_{\lambda B}(u^{-1} - \lambda a^{-1}),$$

where $a^{-1} = A(u^{-1})$. Vector $u^*$ is the solution to problem (22). Hence, there exists $b^* \in B(u^*)$ such that $a^* + b^* = 0$, where $a^* = A(u^*)$. This implies

$$u^* = \mathrm{J}_{\lambda B}(u^* - \lambda a^*).$$

Using the firm non-expansiveness of $\mathrm{J}_{\lambda B}$ we get

$$\|u^0 - u^*\|^2 = \|\mathrm{J}_{\lambda B}(u^{-1} - \lambda a^{-1}) - \mathrm{J}_{\lambda B}(u^* - \lambda a^*)\|^2$$

$$\le \|(u^{-1} - \lambda a^{-1}) - (u^* - \lambda a^*)\|^2$$

$$- \|\mathrm{J}_{\lambda B}(u^{-1} - \lambda a^{-1}) - \mathrm{J}_{\lambda B}(u^* - \lambda a^*) - (u^{-1} - \lambda a^{-1}) + (u^* - \lambda a^*)\|^2$$

$$= \|u^{-1} - u^* - \lambda(a^{-1} - a^*)\|^2 - \|(u^* - \lambda a^* - u^*) - (u^{-1} - \lambda a^{-1} - u^0)\|^2$$

$$= \|u^{-1} - u^* - \lambda(a^{-1} - a^*)\|^2 - \|\lambda a^* + (u^{-1} - \lambda a^{-1} - u^0)\|^2.$$

Using line 5 of Algorithm 3 we get

$$\|u^0 - u^*\|^2 \le \|u^{-1} - u^* - \lambda(a^{-1} - a^*)\|^2 - \|\lambda a^* + \lambda b^0\|^2$$
$$= \|u^{-1} - u^* - \lambda(a^{-1} - a^*)\|^2 - \|\lambda(a^* - a^0) + \lambda(a^0 + b^0)\|^2.$$

Using the inequality $\|\lambda(a^* - a^0) + \lambda(a^0 + b^0)\|^2 \ge \frac{\lambda^2}{2}\|a^0 + b^0\|^2 - \lambda^2\|a^0 - a^*\|^2$ we get

$$\|u^0 - u^*\|^2 \le \|u^{-1} - u^* - \lambda(a^{-1} - a^*)\|^2 - \frac{\lambda^2}{2}\|a^0 + b^0\|^2 + \lambda^2\|a^0 - a^*\|^2.$$

Using the inequality $\|u^{-1} - u^* - \lambda(a^{-1} - a^*)\|^2 \le 2\|u^{-1} - u^*\|^2 + 2\lambda^2\|a^{-1} - a^*\|^2$ we get

$$\|u^0 - u^*\|^2 \le 2\|u^{-1} - u^*\|^2 + 2\lambda^2\|a^{-1} - a^*\|^2 - \frac{\lambda^2}{2}\|a^0 + b^0\|^2 + \lambda^2\|a^0 - a^*\|^2.$$

Using the $M$-Lipschitzness of $A(u)$ we get

$$\|u^0 - u^*\|^2 \le 2\|u^{-1} - u^*\|^2 + 2\lambda^2 M^2\|u^{-1} - u^*\|^2 - \frac{\lambda^2}{2}\|a^0 + b^0\|^2 + \lambda^2 M^2\|u^0 - u^*\|^2.$$

After rearranging we get

$$(1 - \lambda^2 M^2)\|u^0 - u^*\|^2 + \frac{\lambda^2}{2}\|a^0 + b^0\|^2 \le 2(1 + \lambda^2 M^2)\|u^{-1} - u^*\|^2.$$

Plugging the definition of $\lambda$ gives

$$\frac{4}{5}\|u^0 - u^*\|^2 + \frac{1}{10M^2}\|a^0 + b^0\|^2 \le \frac{12}{5}\|u^{-1} - u^*\|^2.$$

Multiplying both sides of the inequality by $120M^2$ gives

$$96M^2\|u^0 - u^*\|^2 + 12\|a^0 + b^0\|^2 \le M^2\|u^{-1} - u^*\|^2.$$

$\square$

**Lemma 9.** *Under conditions of Theorem 2 the following equality holds:*

$$\prod_{t=0}^{T-1}(1 - \beta_t) = \frac{2}{(T+1)(T+2)}. \tag{47}$$

*Proof.*

$$\prod_{t=0}^{T-1}(1 - \beta_t) = \prod_{t=0}^{T-1}\left(1 - \frac{2}{t+3}\right) = \prod_{t=0}^{T-1}\frac{t+1}{t+3} = \frac{\prod_{t=1}^{T} t}{\prod_{t=3}^{T+2} t} = \frac{2}{(T+1)(T+2)}.$$

$\square$

**Lemma 10.** *Under conditions of Theorem 2, let $\mathcal{U}^t$ be the following Lyapunov function*

$$\mathcal{U}^t = \langle a^t + b^t, u^t - u^0 \rangle + \frac{\lambda}{2\beta_t}\|a^t + b^t\|^2. \tag{48}$$

*Then, the following inequality holds for all $t \in \{0, 1, 2, \ldots, T-1\}$:*

$$\mathcal{U}^{t+1} \le (1 - \beta_t)\mathcal{U}^t. \tag{49}$$

*Proof.* We start with the monotonicity property of operators $A(u)$ and $B(u)$:

$$\langle u^{t+1} - u^t, (a^{t+1} + b^{t+1}) - (a^t + b^t) \rangle \ge 0, \tag{50}$$

where $t \in \{0, 1, 2, \ldots, T-1\}$. From line 10 of Algorithm 3 we can conclude that

$$u^{t+1} = u^t + \beta_t(u^0 - u^t) - \lambda(a^{t+1/2} + b^{t+1}), \tag{51}$$

where $a^{t+1/2} = A(u^{t+1/2})$. From this we also conclude that

$$u^{t+1} = u^t + (1 - \beta_t)^{-1} \left[ \beta_t(u^0 - u^{t+1}) - \lambda(a^{t+1/2} + b^{t+1}) \right]. \tag{52}$$

Plugging (51) and (52) into (50) gives

$$0 \le (1 - \beta_t)^{-1} \langle a^{t+1} + b^{t+1}, \beta_t(u^0 - u^{t+1}) - \lambda(a^{t+1/2} + b^{t+1}) \rangle$$
$$- \langle a^t + b^t, \beta_t(u^0 - u^t) - \lambda(a^{t+1/2} + b^{t+1}) \rangle$$
$$= -(1 - \beta_t)^{-1} \left[ \beta_t \langle a^{t+1} + b^{t+1}, u^{t+1} - u^0 \rangle + \lambda \langle a^{t+1} + b^{t+1}, a^{t+1/2} + b^{t+1} \rangle \right]$$
$$+ \beta_t \langle a^t + b^t, u^t - u^0 \rangle + \lambda \langle a^t + b^t, a^{t+1/2} + b^{t+1} \rangle.$$

Using the parallelogram rule we get

$$0 \le -(1 - \beta_t)^{-1} \beta_t \langle a^{t+1} + b^{t+1}, u^{t+1} - u^0 \rangle$$
$$- \frac{(1 - \beta_t)^{-1} \lambda}{2} \left[ \|a^{t+1} + b^{t+1}\|^2 + \|a^{t+1/2} + b^{t+1}\|^2 - \|a^{t+1} - a^{t+1/2}\|^2 \right]$$
$$+ \beta_t \langle a^t + b^t, u^t - u^0 \rangle$$
$$+ \frac{\lambda}{2} \left[ \|a^t + b^t\|^2 + \|a^{t+1/2} + b^{t+1}\|^2 - \|a^t + b^t - (a^{t+1/2} + b^{t+1})\|^2 \right]$$
$$= \beta_t \langle a^t + b^t, u^t - u^0 \rangle + \frac{\lambda}{2} \|a^t + b^t\|^2$$
$$- (1 - \beta_t)^{-1} \left[ \beta_t \langle a^{t+1} + b^{t+1}, u^{t+1} - u^0 \rangle + \frac{\lambda}{2} \|a^{t+1} + b^{t+1}\|^2 \right]$$
$$+ \frac{\lambda(1 - (1 - \beta_t)^{-1})}{2} \|a^{t+1/2} + b^{t+1}\|^2 + \frac{\lambda(1 - \beta_t)^{-1}}{2} \|a^{t+1} - a^{t+1/2}\|^2$$
$$- \frac{\lambda}{2} \|a^t + b^t - (a^{t+1/2} + b^{t+1})\|^2$$
$$= \beta_t \langle a^t + b^t, u^t - u^0 \rangle + \frac{\lambda}{2} \|a^t + b^t\|^2$$
$$- (1 - \beta_t)^{-1} \left[ \beta_t \langle a^{t+1} + b^{t+1}, u^{t+1} - u^0 \rangle + \frac{\lambda}{2} \|a^{t+1} + b^{t+1}\|^2 \right]$$
$$- \frac{\lambda \beta_t(1 - \beta_t)^{-1}}{2} \|a^{t+1/2} + b^{t+1}\|^2 + \frac{\lambda(1 - \beta_t)^{-1}}{2} \|a^{t+1} - a^{t+1/2}\|^2$$
$$- \frac{\lambda}{2} \|a^t + b^t - (a^{t+1/2} + b^{t+1})\|^2.$$

Now, we divide both sides of the inequality by $\beta_t$ and get

$$0 \le \langle a^t + b^t, u^t - u^0 \rangle + \frac{\lambda}{2\beta_t} \|a^t + b^t\|^2$$
$$- (1 - \beta_t)^{-1} \left[ \langle a^{t+1} + b^{t+1}, u^{t+1} - u^0 \rangle + \frac{\lambda}{2\beta_t} \|a^{t+1} + b^{t+1}\|^2 \right]$$
$$- \frac{\lambda}{2(1 - \beta_t)} \|a^{t+1/2} + b^{t+1}\|^2 + \frac{\lambda}{2\beta_t(1 - \beta_t)} \|a^{t+1} - a^{t+1/2}\|^2$$
$$- \frac{\lambda}{2\beta_t} \|a^t + b^t - (a^{t+1/2} + b^{t+1})\|^2.$$

using the inequality $\|a^{t+1/2} + b^{t+1}\|^2 \ge \frac{1}{2}\|a^{t+1} + b^{t+1}\|^2 - \|a^{t+1} - a^{t+1/2}\|^2$ we get

$$0 \le \langle a^t + b^t, u^t - u^0 \rangle + \frac{\lambda}{2\beta_t} \|a^t + b^t\|^2$$
$$- (1 - \beta_t)^{-1} \left[ \langle a^{t+1} + b^{t+1}, u^{t+1} - u^0 \rangle + \frac{\lambda}{2\beta_t} \|a^{t+1} + b^{t+1}\|^2 \right]$$
$$- \frac{\lambda}{4(1 - \beta_t)} \|a^{t+1} + b^{t+1}\|^2 + \frac{\lambda}{2(1 - \beta_t)} \|a^{t+1/2} + a^{t+1}\|^2 + \frac{\lambda}{2\beta_t(1 - \beta_t)} \|a^{t+1} - a^{t+1/2}\|^2$$

$$- \frac{\lambda}{2\beta_t} \|a^t + b^t - (a^{t+1/2} + b^{t+1})\|^2$$

$$= \langle a^t + b^t, u^t - u^0 \rangle + \frac{\lambda}{2\beta_t} \|a^t + b^t\|^2$$

$$- (1 - \beta_t)^{-1} \left[ \langle a^{t+1} + b^{t+1}, u^{t+1} - u^0 \rangle + \frac{\lambda}{2} \left( \frac{1}{\beta_t} + \frac{1}{2} \right) \|a^{t+1} + b^{t+1}\|^2 \right]$$

$$+ \frac{\lambda}{2\beta_t} \left( \frac{1 + \beta_t}{1 - \beta_t} \|a^{t+1/2} + a^{t+1}\|^2 - \|a^t + b^t - (a^{t+1/2} + b^{t+1})\|^2 \right).$$

From the definition of $\beta_t$ it follows that $\frac{1}{\beta_{t+1}} = \frac{1}{\beta_t} + \frac{1}{2}$ and $\frac{1 + \beta_t}{1 - \beta_t} \leq 5$. Hence,

$$0 \leq \langle a^t + b^t, u^t - u^0 \rangle + \frac{\lambda}{2\beta_t} \|a^t + b^t\|^2$$

$$- (1 - \beta_t)^{-1} \left[ \langle a^{t+1} + b^{t+1}, u^{t+1} - u^0 \rangle + \frac{\lambda}{2\beta_{t+1}} \|a^{t+1} + b^{t+1}\|^2 \right]$$

$$+ \frac{\lambda}{2\beta_t} \left( 5\|a^{t+1/2} + a^{t+1}\|^2 - \|a^t + b^t - (a^{t+1/2} + b^{t+1})\|^2 \right).$$

Using the $M$-Lipschitzness of $A(u)$ we get

$$0 \leq \langle a^t + b^t, u^t - u^0 \rangle + \frac{\lambda}{2\beta_t} \|a^t + b^t\|^2$$

$$- (1 - \beta_t)^{-1} \left[ \langle a^{t+1} + b^{t+1}, u^{t+1} - u^0 \rangle + \frac{\lambda}{2\beta_{t+1}} \|a^{t+1} + b^{t+1}\|^2 \right]$$

$$+ \frac{\lambda}{2\beta_t} \left( 5M^2 \|u^{t+1/2} + u^{t+1}\|^2 - \|a^t + b^t - (a^{t+1/2} + b^{t+1})\|^2 \right).$$

From lines 7 and 10 of Algorithm 3 it follows that

$$u^{t+1/2} - u^t = \lambda(a^t + b^t - (a^{t+1/2} + b^{t+1}))$$

Hence,

$$0 \leq \langle a^t + b^t, u^t - u^0 \rangle + \frac{\lambda}{2\beta_t} \|a^t + b^t\|^2$$

$$- (1 - \beta_t)^{-1} \left[ \langle a^{t+1} + b^{t+1}, u^{t+1} - u^0 \rangle + \frac{\lambda}{2\beta_{t+1}} \|a^{t+1} + b^{t+1}\|^2 \right]$$

$$+ \frac{\lambda}{2\beta_t} \left( 5M^2\lambda^2 - 1 \right) \|a^t + b^t - (a^{t+1/2} + b^{t+1})\|^2$$

Using the definition of $\lambda$ we get

$$0 \leq \langle a^t + b^t, u^t - u^0 \rangle + \frac{\lambda}{2\beta_t} \|a^t + b^t\|^2$$

$$- (1 - \beta_t)^{-1} \left[ \langle a^{t+1} + b^{t+1}, u^{t+1} - u^0 \rangle + \frac{\lambda}{2\beta_{t+1}} \|a^{t+1} + b^{t+1}\|^2 \right].$$

Rearranging and multiplying both sides of the inequality by $(1 - \beta_t)$ concludes the proof.

$\square$

Now, we are ready to prove Theorem 2.

**Proof of Theorem 2.** Unrolling the recurrence from Lemma 10 we get

$$\mathcal{U}^T \leq \prod_{t=0}^{T-1} \mathcal{U}^0.$$

Using Lemma 9 we get

$$\mathcal{U}^T \le \frac{2}{(T+1)(T+2)}\mathcal{U}^0.$$

Using the definition of $\mathcal{U}^t$ we get

$$\frac{\lambda}{\beta_0(T+1)(T+2)}\|a^0+b^0\|^2 \ge \langle a^T+b^T, u^T-u^0\rangle + \frac{\lambda}{2\beta_T}\|a^T+b^T\|^2.$$

Using the definition of $\beta_t$ we get

$$\frac{3\lambda}{2(T+1)(T+2)}\|a^0+b^0\|^2 \ge \langle a^T+b^T, u^T-u^0\rangle + \frac{\lambda(T+3)}{4}\|a^T+b^T\|^2.$$

Vector $u^*$ is the solution to problem (22). Hence, there exists $b^* \in B(u^*), a^* = A(u^*)$ such that $a^*+b^* = 0$. From the monotonicity assumption it follows that

$$\langle a^T+b^T, u^T-u^*\rangle = \langle a^T+b^T-(a^*+b^*), u^T-u^*\rangle \ge 0.$$

Hence,

$$\frac{3\lambda}{2(T+1)(T+2)}\|a^0+b^0\|^2 \ge \langle a^T+b^T, u^*-u^0\rangle + \frac{\lambda(T+3)}{4}\|a^T+b^T\|^2.$$

Using the Young's inequality we get

$$\begin{aligned}
\frac{3\lambda}{2(T+1)(T+2)}\|a^0+b^0\|^2 &\ge -\frac{\lambda(T+3)}{8}\|a^T+b^T\|^2 - \frac{2}{\lambda(T+3)}\|u^0-u^*\|^2 \\
&\quad + \frac{\lambda(T+3)}{4}\|a^T+b^T\|^2 \\
&= \frac{\lambda(T+3)}{8}\|a^T+b^T\|^2 - \frac{2}{\lambda(T+3)}\|u^0-u^*\|^2.
\end{aligned}$$

After rearranging we get

$$\frac{\lambda(T+3)}{8}\|a^T+b^T\|^2 \le \frac{2}{\lambda(T+3)}\|u^0-u^*\|^2 + \frac{3\lambda}{2(T+1)(T+2)}\|a^0+b^0\|^2.$$

Multiplying both sides of the inequality by $\frac{8}{\lambda(T+3)}$ gives

$$\|a^T+b^T\|^2 \le \frac{8}{\lambda^2(T+3)^2}\|u^0-u^*\|^2 + \frac{12\lambda}{(T+1)(T+2)(T+3)}\|a^0+b^0\|^2.$$

Plugging the definition of $\lambda$ gives

$$\begin{aligned}
\|a^T+b^T\|^2 &\le \frac{40M^2}{(T+3)^2}\|u^0-u^*\|^2 + \frac{12}{(T+1)(T+2)(T+3)}\|a^0+b^0\|^2 \\
&\le \frac{1}{(T+1)^2}\left(96M^2\|u^0-u^*\|^2 + 12\|a^0+b^0\|^2\right)
\end{aligned}$$

Using Lemma 8 we get

$$\|a^T+b^T\|^2 \le \frac{288M^2}{(T+1)^2}\|u^{-1}-u^*\|^2.$$

$\square$

# F  Proof of Lemma 3

Monotonicity of $A^k(u)$ can be verified trivially. Maximal monotonicity of $B(u)$ follows from the fact that it is equal to the subdifferential of a convex function $(\gamma_x^{-1/2}x, \gamma_y^{-1/2}y) \mapsto r(x) + g(y)$ and (Rockafellar and Wets, 2009, Theorem 12.17).

Now, we prove Lipschitzness of operator $A^k(u)$. Denoting $u_1 = (\gamma_x^{-1/2}x_1, \gamma_y^{-1/2}y_1)$ and $u_2 = (\gamma_x^{-1/2}x_2, \gamma_y^{-1/2}y_2)$ gives

$$\begin{aligned}
\|A^k(u_1) - A^k(u_2)\|^2 &\leq \gamma_x\|a_x^k(x_1, y_1) - a_x^k(x_2, y_2)\|^2 + \gamma_y\|a_y^k(x_1, y_1) - a_y^k(x_2, y_2)\|^2 \\
&\leq 2\gamma_x\|a_x^k(x_1, y_1) - a_x^k(x_1, y_2)\|^2 + 2\gamma_x\|a_x^k(x_1, y_2) - a_x^k(x_2, y_2)\|^2 \\
&\quad + 2\gamma_y\|a_y^k(x_1, y_1) - a_y^k(x_1, y_2)\|^2 + 2\gamma_y\|a_y^k(x_1, y_2) - a_y^k(x_2, y_2)\|^2
\end{aligned}$$

Using the definition of operators $a_x^k(x, y)$ and $a_y^k(x, y)$, the definition of function $\hat{F}(x, y)$ and Assumptions 1 to 3 we get

$$\begin{aligned}
\|A^k(u_1) - A^k(u_2)\|^2 &\leq 2\gamma_x L^2\|y_1 - y_2\|^2 + 2\gamma_x L^2\|x_1 - x_2\|^2 \\
&\quad + 2\gamma_y(L + \theta_y^{-1})^2\|y_1 - y_2\|^2 + 2\gamma_y L^2\|x_1 - x_2\|^2 \\
&= 2(\gamma_x\gamma_y L^2 + \gamma_y^2(L + \theta_y^{-1})^2)\gamma_y^{-1}\|y_1 - y_2\|^2 \\
&\quad + 2(\gamma_x^2 L^2 + \gamma_x\gamma_y L^2)\gamma_x^{-1}\|x_1 - x_2\|^2 \\
&\leq 4\max\{\gamma_x^2 L^2, \gamma_x\gamma_y L^2, \gamma_y^2(L + \theta_y^{-1})^2\}\|u_1 - u_2\|^2 \\
&= 4\max\{\gamma_x^2 L^2, \gamma_y^2(L + \theta_y^{-1})^2\}\|u_1 - u_2\|^2.
\end{aligned}$$

$\square$

# G  Proof of Lemma 4

To prove this lemma it is sufficient to show that condition (31) is satisfied when $t = T$, where $T$ is given by (33).

Let $(x^{k,*}, y^{k,*})$ be the solution of the monotone inclusion problem

$$0 \in A^k((\gamma_x^{-1/2}x^{k,*}, \gamma_y^{-1/2}y^{k,*})) + B((\gamma_x^{-1/2}x^{k,*}\gamma_y^{-1/2}y^{k,*})).$$

Note, that this solution always exists. One can easily show that operator $A^k(u)$ is strongly monotone, which implies the following inequality:

$$\begin{aligned}
\langle x^{k,t} - x^{k,*}, a_x^k(x^{k,t}, y^{k,t}) + b_x^{k,t}\rangle + \langle y^{k,t} - y^{k,*}, a_y^k(x^{k,t}, y^{k,t}) + b_y^{k,t}\rangle &\geq \\
&\geq \frac{\mu_x}{2}\|x^{k,t} - x^{k,*}\|^2 + \theta_y^{-1}\|y^{k,t} - y^{k,*}\|^2 \\
&\geq \gamma_x^{-1}\|x^{k,t} - x^{k,*}\|^2 + \gamma_y^{-1}\|y^{k,t} - y^{k,*}\|^2.
\end{aligned}$$

The latter inequality implies

$$\begin{aligned}
\gamma_x^{-1}\|x^{k,t} - x^{k,*}\|^2 + \gamma_y^{-1}\|y^{k,t} - y^{k,*}\|^2 &\leq \\
&\leq \gamma_x\|a_x^k(x^{k,t}, y^{k,t}) + b_x^{k,t}\|^2 + \gamma_y\|a_y^k(x^{k,t}, y^{k,t}) + b_y^{k,t}\|^2.
\end{aligned}$$

Further, we get

$$\begin{aligned}
\gamma_x^{-1}\|x^{k,-1} - x^{k,*}\|^2 + \gamma_y^{-1}\|y^{k,-1} - y^{k,*}\|^2 & \\
&\leq 2\gamma_x^{-1}\|x^{k,-1} - x^{k,t}\|^2 + 2\gamma_y^{-1}\|y^{k,-1} - y^{k,t}\|^2 \\
&\quad + 2\gamma_x^{-1}\|x^{k,t} - x^{k,t}\|^2 + 2\gamma_y^{-1}\|y^{k,t} - y^{k,t}\|^2 \\
&\leq 2\gamma_x^{-1}\|x^{k,-1} - x^{k,t}\|^2 + 2\gamma_y^{-1}\|y^{k,-1} - y^{k,t}\|^2 \\
&\quad + 2\gamma_x\|a_x^k(x^{k,t}, y^{k,t}) + b_x^{k,t}\|^2 + 2\gamma_y\|a_y^k(x^{k,t}, y^{k,t}) + b_y^{k,t}\|^2.
\end{aligned}$$

Using Theorem 2 and the definition (27) of operators $A^k(u)$ and $B(u)$ we get the following inequality:

$$\gamma_x \|a_x^k(x^{k,t}, y^{k,t}) + b_x^{k,t}\|^2 + \gamma_y \|a_y^k(x^{k,t}, y^{k,t}) + b_y^{k,t}\|^2$$

$$\leq \frac{288M^2}{(t+1)^2} \left( \gamma_x^{-1} \|x^{k,-1} - x^{k,*}\|^2 + \gamma_y^{-1} \|y^{k,-1} - y^{k,*}\|^2 \right)$$

$$\leq \frac{288M^2}{(t+1)^2} \left( 2\gamma_x^{-1} \|x^{k,-1} - x^{k,t}\|^2 + 2\gamma_y^{-1} \|y^{k,-1} - y^{k,t}\|^2 \right)$$

$$+ \frac{288M^2}{(t+1)^2} \left( 2\gamma_x \|a_x^k(x^{k,t}, y^{k,t}) + b_x^{k,t}\|^2 + 2\gamma_y \|a_y^k(x^{k,t}, y^{k,t}) + b_y^{k,t}\|^2 \right)$$

$$= \frac{576M^2}{(t+1)^2} \left( \gamma_x^{-1} \|x^{k,-1} - x^{k,t}\|^2 + \gamma_y^{-1} \|y^{k,-1} - y^{k,t}\|^2 \right)$$

$$+ \frac{576M^2}{(t+1)^2} \left( \gamma_x \|a_x^k(x^{k,t}, y^{k,t}) + b_x^{k,t}\|^2 + s\gamma_y \|a_y^k(x^{k,t}, y^{k,t}) + b_y^{k,t}\|^2 \right)$$

Now, we set $t = T$, where $T$ is defined by (33), and use the definition (30) of $M$. This implies.

$$\gamma_x \|a_x^k(x^{k,t}, y^{k,t}) + b_x^{k,t}\|^2 + \gamma_y \|a_y^k(x^{k,t}, y^{k,t}) + b_y^{k,t}\|^2$$

$$\leq \frac{1}{2} \left( \gamma_x^{-1} \|x^{k,-1} - x^{k,t}\|^2 + \gamma_y^{-1} \|y^{k,-1} - y^{k,t}\|^2 \right)$$

$$+ \frac{1}{2} \left( \gamma_x \|a_x^k(x^{k,t}, y^{k,t}) + b_x^{k,t}\|^2 + \gamma_y \|a_y^k(x^{k,t}, y^{k,t}) + b_y^{k,t}\|^2 \right).$$

After rearranging, we obtain condition (31) with $t = T$.

Now, one can observe that there are two possibilities: the while-loop of Algorithm 4 stops when $t = T$, otherwise it stops earlier. This concludes the proof. □

## H   Proof of Theorem 3

According to Theorem 1, the following number of outer iterations of Algorithm 4 are required to find an $\epsilon$-accurate solution:

$$K = \mathcal{O}\left( \max\left\{ \frac{1}{\alpha}, \frac{\alpha}{\theta_y \mu_y} \right\} \log \frac{1}{\epsilon} \right). \tag{53}$$

Using the definition of $\alpha$ and $\theta_y$ we get

$$K = \mathcal{O}\left( \max\left\{ 1, \sqrt{\frac{\mu_x}{\mu_y}} \right\} \log \frac{1}{\epsilon} \right). \tag{54}$$

According to Lemma 4 at most $T$ inner iterations are performed by Algorithm 4 at each outer iteration, where $T$ is given as

$$T = \mathcal{O}\left( \max\left\{ \frac{L}{\mu_x}, \theta_y L \right\} \right). \tag{55}$$

Using the definition of $\theta_y$ we get

$$T = \mathcal{O}\left( \frac{L}{\mu_x} \right). \tag{56}$$

Now, we get the total number of iterations:

$$K \times T = \mathcal{O}\left( \max\left\{ 1, \sqrt{\frac{\mu_x}{\mu_y}} \right\} \log \frac{1}{\epsilon} \right) \times \mathcal{O}\left( \frac{L}{\mu_x} \right) = \mathcal{O}\left( \max\left\{ \frac{L}{\mu_x}, \frac{L}{\sqrt{\mu_x \mu_y}} \right\} \log \frac{1}{\epsilon} \right). \tag{57}$$

It remains to observe that Algorithm 4 performs $\mathcal{O}(1)$ gradient evaluations per iteration. □