# OpenReview forum: "The First Optimal Algorithm for Smooth and Strongly-Convex-Strongly-Concave Minimax Optimization"
_NeurIPS.cc/2022/Conference — NeurIPS 2022 Accept_

### Official Review · Reviewer_H9Wz · 2022-07-10

**Rating:** 7
**Confidence:** 2
**Soundness:** 3 good
**Presentation:** 3 good
**Contribution:** 3 good

**Summary:**

This paper studies the optimization of the smooth and and strongly-convex-strongly-concave minimax objective. The proposed algorithm meets the previously established lower bound on the number of gradient evaluations required, thereby being the first optimal algorithm for this class of problem.


**Questions:**

N.A.

**Limitations:**

As above.

**Strengths And Weaknesses:**

It is notable that the work of  Wang and Li (2020) is close to optimality with the extra multiplicative term $\log(\kappa_x \kappa_y)^3$, yet having simpler algorithmic structure. On the other hand, the proposed algorithm in this paper has the merit of being the theoretically optimal algorithm, but it can become complicated to implement in practice. It would be nice if the authors can have some experiments just as a simple proof-of-work.

---

> ### Author Response · Authors · 2022-08-01
> **Response to Reviewer H9Wz**
>
> We thank Reviewer H9Wz for the time and effort. We are glad that the reviewer gave such a high estimate of our work. Further, we address the main question raised by the reviewer:
>
> > It is notable that the work of Wang and Li (2020) is close to optimality with the extra multiplicative term $\log(\kappa_x\kappa_y)^3$, yet having simpler algorithmic structure. On the other hand, the proposed algorithm in this paper has the merit of being the theoretically optimal algorithm, but it can become complicated to implement in practice. It would be nice if the authors can have some experiments just as a simple proof-of-work.
>
> The question of the practical efficiency of our algorithm is indeed an important question. However, to the best of our knowledge, the algorithm of Wang and Li (2020), called Proximal Best Response, requires calling APPA-ABR algorithm at each iteration, which in turn requires calling ABR algorithm at each iteration, which in turn requires calling AGD algorithm at each iteration. This results in the 4-level nested structure, which is rather complicated compared to our double-loop algorithm. Another important difference is that our algorithm and the algorithm of Wang and Li (2020) use different stopping criteria. The theory suggests that our stopping criterion is more precise due to the fact that our algorithm does not have additional logarithmic factors in the complexity (such factors typically appear due to stacking multiple algorithms). Hence, we do not expect major issues with the implementation and performance of our algorithm. On the other hand, we agree that it would be nice to perform an experimental comparison of our algorithm with the existing state of the art to verify the practical efficiency of our algorithm. Unfortunately, we can't perform this experiment at the moment due to a short author response period, but we will add such an experiment in the revised version of our paper.

---

### Official Review · Reviewer_scRz · 2022-07-11

**Rating:** 7
**Confidence:** 4
**Soundness:** 4 excellent
**Presentation:** 4 excellent
**Contribution:** 4 excellent

**Summary:**

This paper firstly proposed the optimal first-order algorithm for the smooth and strongly-convex-strongly-concave minimax optimization problem with different parameters of strong convexity and strong concavity.

**Questions:**

See the Weaknesses in "Strengths And Weaknesses".

**Limitations:**

There is no potential negative societal impact of their work.

Please see suggestions in "Strengths And Weaknesses".

**Strengths And Weaknesses:**

Strengths:

This paper proposed an optimal algorithm for strongly-convex-strongly-concave minimax optimization with different parameters of convexity and concavity. The main idea is reformulating the minimax problem by pointwise conjugate function and Moreau-Yosida Regularization. Interestingly, Extra Anchored Gradient method can obtain a sufficient accurate solution of the sub-problem in O(1) complexity. As a result, the total first-order oracle complexity of the algorithm matches the lower bound.

Although all components are well-studied, applying them to establish the optimal algorithm is non-trivial. I believe the theoretical contribution of this paper is valuable.


Weaknesses:
1. The implementation of the proposed algorithm is complicated. It has two-loops and a plenty of parameters. I wonder whether it could perform well in practice.
2. The analysis in this paper only differ the parameters of strong convexity and strong concavity, but the lower bound of Zhang et al. (2021) also consider the different parameters of smoothness. Is the algorithm still optimal in Zhang et al’s more general setting?

---

> ### Author Response · Authors · 2022-08-01
> **Response to Reviewer scRz**
>
> We thank Reviewer scRz for the time and effort. We are glad that the reviewer found our theoretical contributions valuable and non-trivial. Further, we address the main weaknesses raised by the reviewer.
>
> >The implementation of the proposed algorithm is complicated. It has two-loops and a plenty of parameters. I wonder whether it could perform well in practice.
>
> It is indeed a valid concern. However, most of our algorithm parameters have simple formulas based only on the parameters of the problem. In addition, the existing "accelerated" competitors, such as Wang and Li (2020) or Lin et al. (2020), also have loops, but in contrast to them, our inner algorithm has a stopping criterion which is efficient in theory and easy to implement in practice. Overall, in the revised version of the paper, we will try to provide experimental comparison to verify practical efficiency of our algorithm.
>
>
> > The analysis in this paper only differ the parameters of strong convexity and strong concavity, but the lower bound of Zhang et al. (2021) also consider the different parameters of smoothness. Is the algorithm still optimal in Zhang et al’s more general setting?
>
> It is indeed an important question (it was also raised by Reviewer LxoR). One can show that our algorithm can achieve the complexity $\mathcal{O}\left(\frac{\sqrt{L_xL_y} + L_{xy}}{\sqrt{\mu_x\mu_y}}\log \frac{1}{\epsilon}\right)$ in this more general smoothness setting. It can be done via a simple reparametrization trick. Hence, our algorithm matches the existing lower bound as long as $L_xL_y \leq L_{xy}^2$. When the latter condition is not satisfied, our algorithm does not match the existing lower bound. Unfortunately, to the best of our knowledge, no algorithm matches this lower bound even up to extra logarithmic factors for minimax problems with non-bilinear coupling. The existing optimal algorithms (Kovalev et al., 2021) and [A,B], mentioned by Reviewer hsnC, rely on the bilinear coupling assumption. In the analysis, they substantially use the fact that functions $p(x)$ and $q(y)$ (see lines 46-47 of our paper) are isolated from the bilinear coupling part. Overall, we think this is an interesting problem and would like to study it in the future. Another interesting part here is that we could try to improve the lower bound. Note that the existing lower bound is provided in the class of minimax problems with bilinear coupling. Hence, we could try to search for a better lower bound outside of this class, i.e., in the class of minimax problems with non-bilinear coupling.

---

> > ### Comment · Reviewer_scRz · 2022-08-06
> > **Thanks for response.**
> >
> > Thanks the authors' response. I encourage the author present the main result under more general smoothness setting in later version. However, I cannot raise my score since there is no numerical experiment in rebuttal revision.

---

> > > ### Author Response · Authors · 2022-08-07
> > > **Thank you.**
> > >
> > > Dear Reviewer scRz,
> > > Thank you for your feedback. We highly appreciate it.

---

### Official Review · Reviewer_LxoR · 2022-07-12

**Rating:** 7
**Confidence:** 3
**Soundness:** 4 excellent
**Presentation:** 4 excellent
**Contribution:** 4 excellent

**Summary:**

This paper proposes a novel algorithm for the smooth and strongly-convex-strongly-concave minimax problem. The proposed method first reformulates the minimax problem as a particular minimization problem, and then solves the reformulated problem by a variant of the proximal point algorithm. The algorithm computes the proximal operator inexactly by extra anchored gradient method.

**Questions:**

I would appreciate it if authors can discuss whether the proposed algorithm is still optimal in the setting mentioned in the weakness part, and what is the main difficulties in the analysis under such assumption.


**Limitations:**

The proposed algorithm is a double loop algorithm, which may be less practical than single loop algorithms.

**Strengths And Weaknesses:**

Strengths:

1. The studied problem is very important since convex-concave minimax optimization have many applications in machine learning area.
2. The paper is well-written and easy to follow.
3. The technical contribution is solid. The proposed framework is novel and it achieves the optimal gradient complexity.

Weakness:

Notice that Wang and Li (2020) provide a more elaborate analysis on the smoothness. Specifically, they assume the objective function is $(L_x,L_{xy},L_y)$-smooth, rather than use a unified smoothness $L=\max \\{ L_x,L_{xy},L_y \\} $. Whether the proposed method is still optimal under such assumption is unknown, but this is not a critical point to address.

---

> ### Author Response · Authors · 2022-08-01
> **Response to Reviewer LxoR**
>
> We thank Reviewer LxoR for the time and effort. We are pleased that the reviewer found our paper well-written and easy to follow and described our technical contribution as solid. As far as we understand, there is one non-critical concern and one limitation raised by the reviewer, which we further address.
>
>
> > Notice that Wang and Li (2020) provide a more elaborate analysis on the smoothness. Specifically, they assume the objective function is $(L_x,L_{xy},L_y)$-smooth, rather than use a unified smoothness $L = \max\{L_x,L_{xy},L_y\}$. Whether the proposed method is still optimal under such assumption is unknown, but this is not a critical point to address.
>
> > I would appreciate it if authors can discuss whether the proposed algorithm is still optimal in the setting mentioned in the weakness part, and what is the main difficulties in the analysis under such assumption.
>
> It is indeed an important question (it was also raised by Reviewer scRz). One can show that our algorithm can achieve the complexity $\mathcal{O}\left(\frac{\sqrt{L_xL_y} + L_{xy}}{\sqrt{\mu_x\mu_y}}\log \frac{1}{\epsilon}\right)$ in this more general smoothness setting. It can be done via a simple reparametrization trick. Hence, our algorithm matches the existing lower bound as long as $L_xL_y \leq L_{xy}^2$. When the latter condition is not satisfied, our algorithm does not match the existing lower bound. Unfortunately, to the best of our knowledge, no algorithm matches this lower bound even up to extra logarithmic factors for minimax problems with non-bilinear coupling. The existing optimal algorithms (Kovalev et al., 2021) and [A,B], mentioned by Reviewer hsnC, rely on the bilinear coupling assumption. In the analysis, they substantially use the fact that functions $p(x)$ and $q(y)$ (see lines 46-47 of our paper) are isolated from the bilinear coupling part. Overall, we think this is an interesting problem and would like to study it in the future. Another interesting part here is that we could try to improve the lower bound. Note that the existing lower bound is provided in the class of minimax problems with bilinear coupling. Hence, we could try to search for a better lower bound outside of this class, i.e., in the class of minimax problems with non-bilinear coupling.
>
>
> > The proposed algorithm is a double loop algorithm, which may be less practical than single loop algorithms.
>
> It is indeed a valid concern. However, in contrast to the existing "accelerated" algorithms with multiple loops, our inner algorithm has a simple stopping criterion which is efficient in theory. Hence, we do not expect major issues with the implementation and performance of our algorithm in practice. In the revised version of our paper, we will try to provide an experimental comparison of our method with state-of-the-art algorithms to verify the practical efficiency of our algorithm.

---

> > ### Comment · Reviewer_LxoR · 2022-08-07
> > **Thanks for response**
> >
> > I appreciate authors' responses to my questions. Overall, I think this paper is an excellent theoretical paper. So I raise my score to 7.

---

> > > ### Author Response · Authors · 2022-08-07
> > > **Thank you.**
> > >
> > > Dear Reviewer LxoR,
> > > Thank you for the high estimate of our work. We highly value your feedback.

---

### Official Review · Reviewer_hsnC · 2022-07-18

**Rating:** 7
**Confidence:** 4
**Soundness:** 3 good
**Presentation:** 2 fair
**Contribution:** 3 good

**Summary:**

Proposes algorithm to solve smooth and strongly-convex-strongly-concave minimax problem of the form $\min_x \min_y r(x) + F(x, y) - h(y)$. Here $r$ and $h$ are proximable and $F$ is differentiable, $L$ smooth and strongly-convex-strongly-concave. The main idea is to convert the minimax problem into an equivalent strongly convex minimization problem and then solve it by applying accelerated GD on it’s Moreau envelope. Gradient of the Moreau envelope is approximated using a new generalization of the recently proposed Extra Anchored Gradient method for making gradients of minimax objectives small.

**Questions:**

1. The title and message of the paper is misleading and exaggerated because they ignore the fact that there exist another very common parameterization of the same minimax problem which allows for different Lipschitz smoothness parameter in $xx$, $yy$ and $xy$ blocks. Most of the papers considered in the Table 1 consider this parameterization. Further, under this parameterization the proposed method is not optimal. I definitely got confused while reading the paper. I recommend the authors to reconsider their message and the title to avoid any misconceptions, and appropriately discuss these other parameterizations.
  - one of the “holy grails” of smooth minimax optimization is the recovery of optimal convex rates in the limit $L_{xy} \to 0$, $L_{yy} \to 0$ when $L_{yy}/\mu_{yy} = 1$. This is not possible in the stricter parameterization studied in this paper. Please discuss this as a limitation.

2. “open question was answered positively in the work of Kovalev et al. (2021) in the case of minimax problems with bilinear coupling”: There are two other papers also which solve this same problem [A, B]. These papers are particularly related and may be more closer to this paper than Kovalev et al. (2021) because the reformulation in eqns (7-8) are similar to [Eqn 14, A] and [Eqn 13, B]. The main idea of this paper to work in the dual space of either the $x$ or the $y$ variable is related to the dualization done in [A, B]. Please appropriately give credit and compare your ideas to these methods.

3. In the proof of Theorem 1, LHS of the final result (eg: initial sub-optimality gap in P(z, y)) is never fully computed in terms of the original problem parameters, gaps and distances. This may end up being a minor issue, but please make your proof more rigorous.

4. Abstract is hand-wavy when stating the complexity is $O(\sqrt{\kappa_x \kappa_y} \log(1/\epsilon))$. Checking the proof of Theorem 1, some $\log(\mu_x)$ and $\log(\mu_y)$ terms are missing. Please make the results more rigorous.

5. There will be numerical instability because of $\mu_x^{-1} = \theta_x^{-1}$ when $\mu_x$ is very small. Does the authors have a solution for this issue? Note that (proximal) mirror-prox can achieve a sub-optimal linear convergence rate using a stable stepsize=4/L, without any numerical instability. Please discuss this limitation in the paper.

6. Out of curiosity, is it possible to use a similar formulation as (13 ) when the minimax objective is not strongly monotone on one of the sides? Currently the reformulation only works for strongly-convex-strongly-convex problems.

[A] Thekumparampil et al., Lifted Primal-Dual Method for Bilinearly Coupled Smooth Minimax Optimization. https://arxiv.org/abs/2201.07427 (also at AISTATS 2022)

[B] Jin et al., Sharper Rates for Separable Minimax and Finite Sum Optimization via Primal-Dual Extragradient Methods. https://arxiv.org/abs/2202.04640 (also at COLT 2022)

**Limitations:**

Limitations were not discussed. See above for ideas.

**Strengths And Weaknesses:**

### Pros
1. Under the assumptions of the paper, the algorithm is better than baselines because they tighten the cubic- or square-log factors in the complexity to just log factor.
2. Great presentation of the main ideas of the algorithm and proof sketch of Algorithm 1.
3. The reformulation of strongly-convex-strongly-concave minimax problem to strongly-convex minimization seems novel and could be potentially useful elsewhere too. However, there are concerns about missing related work.
4. Problem formulation is general. So it can capture both constrained and unconstrained problems, however the paper doesn’t discuss it as such.


### Potential for improvement
1. The title and message of the paper is misleading and exaggerated because they ignore the fact that there exist another very common parameterization of the same minimax problem which allows for different Lipschitz smoothness parameter in $xx$, $yy$ and $xy$ blocks. See below.

2. Can greatly improve the related works section. To give some examples, please discuss papers which separately consider $L_x$, $L_y$ and $L_{xy}$ Lipschitz constants. Also two very related works on bilinear coupling (see [A, B] below) are missing and they need to be discussed.

3. Proof and results are not fully rigorous. See below.

4. There will be numerical instability because of $\mu_x^{-1}$ when $\mu_x$ is very small. See below.

---

> ### Author Response · Authors · 2022-08-01
> **Response to Reviewer hsnC (part 1)**
>
> We thank Reviewer hsnC for the time and effort. We are glad that the reviewer appreciated our theoretical results. Further, we provide detailed comments on all the questions and concerns raised by the reviewer. We do this in separate posts because we run out of character limit.
>
> > The title and message of the paper is misleading and exaggerated because they ignore the fact that there exist another very common parameterization of the same minimax problem which allows for different Lipschitz smoothness parameter in $xx$, $xy$ and $xy$ blocks. Most of the papers considered in the Table 1 consider this parameterization. Further, under this parameterization the proposed method is not optimal. I definitely got confused while reading the paper. I recommend the authors to reconsider their message and the title to avoid any misconceptions, and appropriately discuss these other parameterizations.
>
> We thank Reviewer hsnC for pointing to this. We agree that the title of our paper is a bit exaggerated. In the revised version of the paper, we will provide a detailed discussion of the setting with different Lipschitz constants $L_x$, $L_y$ and $L_{xy}$, and try to provide a more precise title.
>
> > one of the “holy grails” of smooth minimax optimization is the recovery of optimal convex rates in the limit $L_{xy} \rightarrow 0$, $L_{yy} \rightarrow 0$ when $L_{y}/\mu_y = 1$. This is not possible in the stricter parameterization studied in this paper. Please discuss this as a limitation.
>
> This is a valid concern. However, one can show that with a simple reparametrization trick, our algorithm can achieve the convergence rate proportional to $\mathcal{O}\left(\frac{\sqrt{L_xL_y} + L_{xy}}{\sqrt{\mu_x\mu_y}} \log \frac{C}{\epsilon}\right)$. Hence, one can observe that in the limit  $L_{xy} \rightarrow 0$ and when $\frac{L_y}{\mu_y} = \mathcal{O}(1)$, our algorithm achives the convergence rate proportional to $\mathcal{O}\left(\frac{\sqrt{L_x}}{\sqrt{\mu_x}} \log \frac{C}{\epsilon}\right)$, which recovers the optimal convergence rate of Accelerated Gradient Descent for smooth and strongly convex optimization. We will add this discussion in the revised version of our paper.
>
> > “open question was answered positively in the work of Kovalev et al. (2021) in the case of minimax problems with bilinear coupling”: There are two other papers also which solve this same problem [A, B]. These papers are particularly related and may be more closer to this paper than Kovalev et al. (2021) because the reformulation in eqns (7-8) are similar to [Eqn 14, A] and [Eqn 13, B]. The main idea of this paper to work in the dual space of either the $x$ or the
> $y$ variable is related to the dualization done in [A, B]. Please appropriately give credit and compare your ideas to these methods.
>
> We thank Reviewer hsnC for mentioning these references. We will add them and provide a discussion in the revised version of the paper.
>
> >In the proof of Theorem 1, LHS of the final result (eg: initial sub-optimality gap in P(z, y)) is never fully computed in terms of the original problem parameters, gaps and distances. This may end up being a minor issue, but please make your proof more rigorous.
>
> We did not consider this as a major issue because anyway the initial suboptimality gap appears in the additive term under the logarithm in the complexity, which does not depend on $\epsilon$. One possible solution is to make a single step of our algorithms with the parameter $\alpha = 1$. In this case we will be able to bound the functional suboptimality with respect to $P(z,y)$ using initial distances only: see line 401 (end of page 18), choosing $\alpha=1$ cancels the functional suboptimality and we can bound the Lyapunov function $\mathcal{L}^1$ using the initial distances only. Hence, we can bound $P(z_f^1,y_f^1) - P(z^*,y^*)$ using the initial distances as well. After this, we can run our algorithm as usual starting from the obtained points $z^1,z_f^1,y^1,y_f^1$. We thank Reviewer hsnC for drawing our attention to this.

---

> > ### Comment · Reviewer_hsnC · 2022-08-08
> > **Follow up**
> >
> > Thank you for your clarifying answers. These are all great points! Please add it to the next revision. Let me ask 2 follow up questions.
> >
> > 1. Why wasn't the paper revised during the rebuttal? I was looking forward to seeing the promised change in the papers.
> >
> > 2.
> >
> > > We will add them and provide a discussion in the revised version of the paper.
> >
> > For this question, no further discussion is provided. I would really appreciate if the authors can provide the comparison and discussion, and so that I can better understand the relation to these papers.

---

> > > ### Author Response · Authors · 2022-08-09
> > > **Follow up question 1**
> > >
> > > Dear Reviewer hsnC,
> > >
> > > Thank you for your feedback. We are glad that you appreciated our answers. Unfortunately, we could not revise the paper and apply the changes during the rebuttal period because of the following two reasons.
> > >
> > > The first reason is the time limitation. A naive approach would be to put all the explanations we made in the appendix of the paper, but this is probably not what you would like to see. However, incorporating these changes into the main text will take some time to avoid hurting the main text flow. Unfortunately, it is hard to do so within a single week especially taking into account that we have a few more submissions that need to be taken care of.
> > >
> > > The second reason is the page limitation. Unfortunately, at this moment, adding even a single text line to the main paper violates the page limit, which causes difficulties in implementing the changes. But if our paper is accepted, we will be able to use an extra page which will greatly simplify the implementation of the changes.
> > >
> > > Further, we provide an answer to the second question. We will do so in a separate post because we run out of the character limit. We will add the following discussion in the revised version of our paper. Unfortunately, it will be hard to put the whole discussion in the main part due to the page limit, and we will likely place most of it in the appendix.

---

> > > ### Author Response · Authors · 2022-08-09
> > > **Follow up question 2**
> > >
> > > Indeed, both papers (Thekumparampil et al., 2022) and (Jin et al., 2022) use Fenchel conjugate functions $p^*(x)$ and $q^*(y)$ to solve the minimax optimization problem with bilinear coupling
> > > $$
> > > \min_x \max_y p(x) + x^\top A y - q(y).
> > > $$
> > > However, the ideas used in these papers are substantially different from ours and can't be applied to solve the general minimax optimization problem. Let us further clarify this.
> > >
> > > Firstly, to the best of our knowledge, Thekumparampil et al. and Jin et al. use the dual functions $p^*(x)$ and $q^*(y)$ for a single purpose only: Nesterov acceleration. In the paper *"An optimal randomized incremental gradient method"* by Lan and Zhou, it was shown that a variant of accelerated gradient descent for solving the smooth convex minimization problem
> > > $$
> > > \min_x \frac{\mu}{2}\lVert x \rVert^2 + f(x)
> > > $$
> > > can be seen as a variant of Chambolle and Pock's primal-dual method generalized to the relative smoothness setting and applied to the following minimax reformulation:
> > > $$
> > > \min_x\max_y \frac{\mu}{2}\lVert x \rVert^2 + x^\top y - f^*(y).
> > > $$
> > > In order to understand this, one needs to replace the dual proximal step
> > > $$
> > > y^{k+1} = \arg\min_{y} \eta_y f^*(y) + \frac{1}{2} \lVert y - y^k\rVert^2
> > > $$
> > > with the following proximal step
> > > $$
> > > y^{k+1} = \arg\min_{y} \eta_y f^*(y) + D_{f^*}(y, y^k),
> > > $$
> > > where we replaced the standard Euclidean distance with the Bregman divergence associated with the function $f^*(y)$, and $\eta_y > 0$ is the stepsize. One can show that the latter proximal operator can be computed efficiently using computations of $\nabla f(x)$ only, and that the resulting variant of Chambolle and Pock's method has the correct accelerated rate of Nesterov Accelerated Gradient. Thekumparampil et al. refer to the work of Lan and Zhou directly, and Jin et al. refer to the paper *"Relative lipschitzness in extragradient methods and a direct recipe for acceleration"* of Cohen et al., which is based on the same idea applied to the Mirror-prox algorithm instead of Chambolle and Pock's method.
> > >
> > > Since Thekumparampil et al. and Jin et al. use the dual functions $p^*(x)$ and $q^*(y)$ for Nesterov acceleration only, it is possible to provide variants of their algorithms without any duality. For instance, an attempt to make a dual-free variant of the algorithm by Thekumparampil et al. would result in an algorithm that is very similar to the algorithm of Kovalev et al. It is also possible to make a dual-free variant of the algorithm by Jin et al., although it would result in a new dual-free algorithm which does not look like any existing method (to the best of our knowledge). What we are trying to explain here is **that duality does not play a critical role in the results of Thekumparampil et al. and Jin et al.** However, duality plays a critical role in our paper because it allows us to reformulate the minimax optimization problem $\min_x \max_y F(x,y)$ as a minimization problem (10) and apply a variant of Nesterov acceleration to it.
> > >
> > > The algorithms of Thekumparampil et al. and Jin et al. are able to achieve the optimal rates due to the structure of the problem
> > > $
> > > \min_x \max_y p(x) + x^\top A y - q(y).
> > > $
> > > In this problem, the bilinear coupling function $x^\top A y$ is separated from the functions $p(x)$ and $q(x)$ which makes it possible to use minimization algorithms for solving this problem almost directly. Indeed, one can apply an accelerated version of the forward-backward algorithm to the monotone inclusion problem
> > > $$
> > > \text{find } (x^*,y^*) \text{ such that } G(x^*,y^*) + H(x^*,y^*)= 0,
> > > $$
> > > where the monotone operators are defined as $G(x,y) = (\nabla p(x),\nabla q(y))$ and $H(x,y) = (Ay,-A^\top x)$. Indeed, Nesterov acceleration can be applied to the monotone inclusion problem because operator G(x,y) is potential, i.e., it is equal to the gradient of the convex function $(x,y) \mapsto p(x) + q(y)$ (it is explained by Kovalev et al.). The only remaining question is how to compute the backward step with respect to $H(x,y)$. Thekumparampil et al. and Kovalev et al. use the linear extrapolation step of Chambolle and Pock, and Jin et al. use the Extragradient/mirror-prox method for this purpose. Unfortunately, these ideas cannot be used for solving the general minimax optimization problem $\min_x \max_y F(x,y)$. In particular, the analog of backward step cannot be used because it would immediately result in the non-accelerated term $(\frac{L_x}{\mu_x} + \frac{L_y}{\mu_y})\log \frac{1}{\epsilon}$ in the complexity, and Nesterov acceleration cannot be applied because the potential functions $p(x)$ and $q(y)$ are "hidden inside" the function $F(x,y)$. This is where the pointwise conjugate function comes to our rescue, as it allows us to reformulate the minimax problem as the minimization problem (10) and to apply a variant of Nesterov acceleration to it.

---

> ### Author Response · Authors · 2022-08-01
> **Response to Reviewer hsnC (part 2)**
>
> > Abstract is hand-wavy when stating the complexity is $\mathcal{O}(\sqrt{\kappa_x\kappa_y} \log (1/\epsilon))$. Checking the proof of Theorem 1, some $\log(\mu_x)$ and $\log(\mu_y)$ terms are missing. Please make the results more rigorous.
>
> Our theory implies that the complexity of our algorithm is $\mathcal{O}(\sqrt{\kappa_x\kappa_y} \log (C/\epsilon))$, where $C$ is a constant which indeed depends on the parameters of the problem, including $\mu_x$ and $\mu_y$, but does not depend on $\epsilon$. Our complexity can be written as $\mathcal{O}(\sqrt{\kappa_x\kappa_y} \log (1/\epsilon) + \sqrt{\kappa_x\kappa_y}\log C)$. When we wrote the result in the form $\mathcal{O}(\sqrt{\kappa_x\kappa_y} \log (1/\epsilon))$ we meant that $\mathcal{O(\cdot)}$ also hides non-leading terms, and the leading term here is $\sqrt{\kappa_x\kappa_y} \log (1/\epsilon)$ which is the only term depending on $\epsilon$. Please note that such a simplification is also done in abstracts of other works like Wang and Li (2020) or Zhang et al. (2021), who only write terms proportional to $\log (1/\epsilon)$  in the abstract. We agree that this is a bit inaccurate and we will provide a clarification in the revised version of the paper.
>
> > There will be numerical instability because of $\mu_x^{-1} = \theta_x^{-1}$
>  when $\mu_x$ is very small. Does the authors have a solution for this issue? Note that (proximal) mirror-prox can achieve a sub-optimal linear convergence rate using a stable stepsize=$4/L$, without any numerical instability. Please discuss this limitation in the paper.
>
>
> We thank Reviewer hsnC for asking this question. Our algorithm is used in the case $\mu_x > \mu_y$, otherwise we just swap variables $x$ and $y$. This is mentioned in Corollary 1. Hence, the scenario $\mu_x \rightarrow 0$ is unlikely to happen. We will discuss this thoroughly in the revised version of our paper.
>
> > Out of curiosity, is it possible to use a similar formulation as (13 ) when the minimax objective is not strongly monotone on one of the sides? Currently the reformulation only works for strongly-convex-strongly-convex problems.
>
> It is an interesting question. One straightforward approach is to use the regularization trick, following Lin et al. (2020). It will allow us to reach the near-optimal rates. However, if we want to directly apply reformulation (13) in the non-strongly convex case, we do not see any issues with this. Indeed, for any proper closed convex function $f(x)$, the Moreau envelope $f^\gamma(x)$ will have the same set of minimizers as $f(x)$ (see Example 1.46 in Rockafellar, Wets. *Variational Analysis*). From the algorithmic perspective, we think that it is possible to provide a direct variant of our method in the case when the strong convexity is not assumed. In order to do this, we probably need to generalize Algorithm 2 to a non-strongly convex case which is possible. In this scenario, it is unlikely that the analysis of Algorithm 3 should be changed, but it should be checked carefully. We are also not sure whether it will be possible to reach optimal rates with this approach, it should be further investigated. Overall, it is an interesting direction for future work.

---

### Author Response · Authors · 2022-08-01
**Message to all Reviewers and the AC**

Dear reviewers and AC,

We thank you for your time and effort in reading and assessing our paper.
As far as we can understand from the reviews and scores, all the reviewers have reached a consensus and positively evaluated our paper, which we were glad to see.

At the same time, the reviewers made some valuable comments and raised interesting questions. We address these comments and questions in separate posts. We kindly ask the reviewers to let us know if we did not address some of the concerns properly or if new questions have appeared.

---

### Meta-Review · Area_Chair_Vdq9 · 2022-08-29

**Recommendation:** Accept
**Confidence:** Certain

**Metareview:**

The reviewers all agree that the paper considers an important problem, and the results are novel and interesting. Congratulations!

**Award:**

No

---

### Decision · Program_Chairs · 2022-09-14

Accept